# Gap-Aware Mitigation of Gradient Staleness

**Saar Barkai**[*]
Department of Electrical Engineering
Technion - Israel Institute of Technology
Haifa, Israel
saarbarkai@gmail.com

**Ido Hakimi**[*]
Department of Computer Science
Technion - Israel Institute of Technology
Haifa, Israel
idohakimi@gmail.com

**Assaf Schuster**
Department of Computer Science
Technion - Israel Institute of Technology
Haifa, Israel
assaf@sc.technion.ac.il

## Abstract

Cloud computing is becoming increasingly popular as a platform for distributed training of deep neural networks. Synchronous stochastic gradient descent (SSGD) suffers from substantial slowdowns due to stragglers if the environment is non-dedicated, as is common in cloud computing. Asynchronous SGD (ASGD) methods are immune to these slowdowns but are scarcely used due to gradient staleness, which encumbers the convergence process. Recent techniques have had limited success mitigating the gradient staleness when scaling up to many workers (computing nodes). In this paper we define the *Gap* as a measure of gradient staleness and propose Gap-Aware (GA), a novel asynchronous-distributed method that penalizes stale gradients linearly to the Gap and performs well even when scaling to large numbers of workers. Our evaluation on the CIFAR, ImageNet, and WikiText-103 datasets shows that GA outperforms the currently acceptable gradient penalization method, in final test accuracy. We also provide convergence rate proof for GA. Despite prior beliefs, we show that if GA is applied, momentum becomes beneficial in asynchronous environments, even when the number of workers scales up.

## 1 Introduction

The steady growth of deep neural networks over the years has made it impractical to train them from scratch on a single *worker* (i.e., computational device). Distributing the computations over several workers can drastically reduce the training time. However, due to the sequential nature of the widely used stochastic gradient descent (SGD) method, distributing the process is not an easy task.

Synchronous SGD (SSGD) is the most common method used to distribute the learning process across multiple workers. Several recent works (Mikami et al., 2018; Ying et al., 2018; Yamazaki et al., 2019; Goyal et al., 2017) have shown that SSGD can achieve large speedups while maintaining high accuracy. The major drawback of SSGD is that its speed is confined to the slowest worker in every iteration. This shortcoming is magnified in non-dedicated[1] environments such as cloud computing. For this reason, all the above mentioned works were forced to use homogeneous workers in a dedicated network, which serves to reduce the variance in the workers' iteration times. Unlike cloud computing, dedicated networks are expensive and therefore not available to most users.

In asynchronous SGD (ASGD), each worker communicates independently of the others, thereby addressing the major drawback of SSGD. ASGD enjoys linear speedup in terms of the number of workers, even on non-dedicated networks. This makes ASGD a potentially better alternative to SSGD when using cloud computing. Unfortunately, ASGD also has a significant weakness known as *gradient staleness*; the gradients used to update the parameter server's (master) parameters are

---

[*]Equal contribution.
[1]An environment of computation nodes who are not specifically optimized to work together

often based on older parameters and therefore are inaccurate. Prior works have shown that gradient staleness severely hinders the convergence process by reaching reduced final accuracy (Chen et al., 2016; Cui et al., 2016). Mitliagkas et al. (2016) showed that gradient staleness also induces *implicit momentum*, thus the momentum coefficient $\gamma$ must be decayed when scaling up the number of workers. Most research works measure the gradient staleness of a worker according to the *delay*: the number of master updates since the worker began calculating the stochastic gradient $g$, until $g$ is used to update the master. To overcome gradient staleness, Zhang et al. (2015b) proposed *Staleness-Aware* (SA), which penalizes the step size of stale gradients linearly to their *delay*. This method was later embraced by other works (Jiang et al., 2017; Hardy et al., 2017; Giladi et al., 2019) and is currently the common method for penalizing stale gradients. Unfortunately, this method suffers from a degradation of the final accuracy, especially when scaling up the number of workers. In Section 4.1, we show that the main reason for this degradation is the over-penalization and under-penalization caused by SA.

SSGD and ASGD rely on hyperparameter tuning for every different number of workers (Shallue et al., 2018). Tuning is extremely time-consuming, thus avoiding it is beneficial, whenever possible.

**Our contribution:** To mitigate gradient staleness while minimizing the degradation of final accuracy, we define a measure of gradient staleness we refer to as the *Gap*. The Gap is based on the difference between the parameters used to calculate the gradient and the parameters on which the gradient is applied. We propose a new method called *Gap-Aware* (GA) that penalizes the step size of stale gradients linearly to their Gap, while eliminating the over-penalization or under-penalization of SA. No new hyperparameters are introduced using the GA method.

- We show that GA out-performs SA, especially as the number of workers scales up.
- We prove that the convergence rate of the GA-ASGD algorithm with a non-convex loss function is consistent with SGD: $\mathcal{O}\left(\frac{1}{\sqrt{BK}}\right)$ where $K$ is the total number of steps and B is the batch size.
- We show that penalizing the gradient itself rather than the step-size, eliminates under-penalization.
- Our results suggest that GA can be used without re-tuning the hyperparameters.
- As opposed to conclusions by Mitliagkas et al. (2016), we show that applying momentum in an asynchronous environment is advantageous (using GA), even when multiple workers are used.
- We combine GA with Adam (Kingma & Ba, 2015) (Adam-GA), and show that Adam-GA achieves almost two orders of magnitude better perplexity than Adam or Adam-SA (which combines Adam with SA) using several workers on the Transformers-XL model.

Our results establish GA as a superior gradient-penalizing option to SA and suggest that using GA is a preferable alternative to SSGD in non-dedicated networks such as cloud computing, even when scaling to large numbers of workers. To validate our claims, we performed experiments on the CIFAR10, CIFAR100 (Krizhevsky, 2012), ImageNet (Russakovsky et al., 2015), and WikiText-103 (Merity et al., 2016) datasets, using several state-of-the-art architectures. A version of GA has reached 72.18% final test accuracy on the ImageNet dataset using 128 simulated asynchronous workers. As far as we know, this is the largest number of asynchronous workers reported to converge on ImageNet.

## 2  RELATED WORK

Eliminating gradient staleness is a challenging task and several papers suggested techniques to reduce its detrimental effects. Zheng et al. (2017) proposed DC-ASGD, which uses a Taylor expansion to mitigate the gradient staleness. EASGD (Zhang et al., 2015a) uses a *center force* to pull the workers' parameters toward the master's parameters. Both DC-ASGD and EASGD achieve high accuracy on small numbers of workers, but fall short when trained on large clusters. Chan & Lane (2014) proposed penalizing stale gradients by reducing their size and thus limiting their effect on the learning process. They suggest decaying the learning rate exponentially to the *delay*; this makes the step size arbitrarily small when the amount of workers grows, virtually ceasing the learning process. As part of their convergence analysis, Dutta et al. (2018) suggest a penalizing method that is linear to the norm between the master and worker's parameters. However, their method introduces another hyperparameter, which requires additional time to tune. As opposed to other methods, GA performs well even when the number of workers is large, without introducing new hyperparameters.

Gupta et al. (2016) as well as Aji & Heafield (2019) suggest collecting several gradients before updating the master to reduce the effects of gradient staleness. Wen et al. (2017) propose minimizing the size of the gradients to reduce communication times. GA is orthogonal to both of these approaches.

## 3 BACKGROUND

The goal of an optimization procedure is to minimize $f(\theta)$, where $f$ is a smooth, but not necessarily convex, objective function (a.k.a. loss) and the vector $\theta \in \mathbb{R}^d$ is the model's parameters:

$$\theta_* = \arg\min_{\theta \in \mathbb{R}^d} f(\theta) := \mathbb{E}_\xi[F(\theta; \xi)] \tag{1}$$

where $\xi \in \Xi$ is a random variable from $\Xi$, the entire set of training samples $\Xi = \{1, 2, \cdots, M\}$. $F(\theta; \xi)$ is the stochastic loss function with respect to the training sample indexed by $\xi$. SGD is commonly the workhorse behind the optimization of deep neural networks. Denoting $\eta$ as the learning rate and $k$ as the step number, SGD's iterative update rule is: $\theta_{k+1} = \theta_k - \eta_k \nabla f(\theta_k)$. We denote $X_k$ as the variable $X$ at the $k^{th}$ step, where $X$ is any variable.

**Momentum**  Momentum (Polyak, 1964) is a widely adopted optimization technique due to its accelerated convergence and oscillation reduction (Sutskever et al., 2013). Instead of simply using the gradient, the momentum iterative update rule[2] uses an exponentially-weighted moving average of gradients called the *update vector*: $v_{k+1} = \gamma v_k + \nabla f(\theta_k)$. The update rule is: $\theta_{k+1} = \theta_k - \eta_k v_{k+1}$. *Nesterov's Accelerated Gradient* (NAG) (Nesterov, 1983) is a well-used variation of momentum that has been proven to achieve quadratic speedup in convergence rate compared to SGD.

## 4 ASYNCHRONOUS SGD (ASGD)

We consider the commonly used ASGD, which operates with a parameter-server (master), used to keep the model's most up-to-date parameters. Each worker maintains a replica of the model's parameters. The workers run in parallel and synchronize with the master independently from each other at the beginning of each batch iteration. We denote $\tau_k$ as the delay at the $k^{th}$ step. The worker and master algorithms are given by Algorithm 1 and 2, respectively, where B is the batch-size and $\xi_{k,b}$ denotes the $b^{th}$ sample in the batch sampled at the $k^{th}$ iteration. *Gradient staleness* appears when a gradient $g_k$ is computed on parameters $\theta_{k-\tau_k}$ but applied to different parameters $\theta_k$.

| **Algorithm 1** Momentum-ASGD: worker $i$ | **Algorithm 2** Momentum-ASGD: master |
|---|---|
| Always do: | For k = 1...K do: |
| $\quad$ Receive parameters $\theta_{k-\tau_k}$ from the master | $\quad$ Receive gradient $g_k^i$ from worker $i$ |
| $\quad$ Get $B$ training samples $\xi_{k-\tau_k, [1...B]}$ | $\quad$ Update momentum $v_{k+1} \leftarrow \gamma v_k + g_k^i$ |
| $\quad$ Compute gradient: $g_k^i \leftarrow \sum_{b=1}^{B} \frac{\nabla F(\theta_{k-\tau_k}; \xi_{k-\tau_k, b})}{B}$ | $\quad$ Update master's weights: |
| $\quad$ Send $g_k^i$ to the master | $\quad\quad \theta_{k+1} \leftarrow \theta_k - \eta_k \cdot v_{k+1}$ |
| | $\quad$ Send $\theta_{k+1}$ to worker $i$ |

### 4.1 STALENESS-AWARE

Zhang et al. (2015b) proposed a gradient penalization method called *Staleness-Aware* (SA). SA aims to reduce the effects of *gradient staleness* by dividing the step size by its corresponding $\tau$ (Algorithm 3). The worker algorithm remains unchanged. The two ideas behind SA is that stale gradients should be penalized to reduce their impact and that gradient staleness scales up with $\tau$. SA successfully mitigates the gradient staleness when $N$ is small. Although commonly used, SA can potentially over-penalize as well as under-penalize stale gradients as we show below.

---

[2]We consider the version of momentum without dampening

| **Algorithm 3** Staleness-Aware: master | **Algorithm 4** Gap-Aware: master |
|---|---|
| Initialize an iteration array: $iter = [0] * N$ 
 For k = 1...K do: 
      Receive gradient $g_k^i$ from worker $i$ 
      Calculate worker $i$'s delay $\tau_k \leftarrow k - iter[i]$ 
      Update momentum $v_{k+1} \leftarrow \gamma v_k + g_k^i$ 
      Update master $\theta_{k+1} \leftarrow \theta_k - \frac{\eta_k}{\tau_k} v_{k+1}$ 
      Send $\theta_{k+1}$ to worker $i$ 
      Save current iteration $iter[i] \leftarrow k$ | For k = 1...K do: 
      Receive gradient $g_k^i$ from worker $i$ 
      Calculate Gap: $G_k = \frac{|\theta_k - \theta_{k-\tau_k}|}{C} + \mathbf{1}^d$ 
      Update momentum $v_{k+1} \leftarrow \gamma v_k + \left( \frac{1}{G_k} \right) \odot g_k^i$ 
      Update master $\theta_{k+1} \leftarrow \theta_k - \eta_k v_{k+1}$ 
      Save and send current parameters 
          $\theta_{k+1}$ to worker $i$ |

**Over-Penalizing** Let us assume that at some step k, after $\tau$ master updates we get $\theta_k = \theta_{k-\tau}$ just as the gradient calculated on $\theta_{k-\tau}$ is applied. This means there is no gradient staleness for the next gradient update since it was computed using the same parameters on which it is applied. Unfortunately, the delay remains $\tau > 0$, thereby causing over-penalization when SA is used.

Additionally, $\tau$ scales linearly with $N$, which dramatically reduces $\eta$ when $N$ is large. Consequently, on large numbers of workers, the convergence rate of SA is sluggish and its accuracy plummets.

**Under-Penalizing** The SA method doesn't take into account that when using momentum, the update step also contains past gradients. To emphasize the importance of this issue, let's examine a fictional example: assume that some gradient $g_k$ is very stale ($\tau_k$ is large). Following the SA technique, the update rule is: $\theta_{k+1} = \theta_k - \frac{\eta_k}{\tau_k} \cdot v_{k+1} = \theta_k - \frac{\eta_k}{\tau_k} \cdot (\gamma v_k + g_k)$. This means the stale gradient $g_k$ is indeed penalized by being multiplied by $\frac{\eta_k}{\tau_k}$, which is small; thus $g_k$ doesn't change the parameters much. We further assume the next iteration is very fast ($\tau_{k+1}$ is small). The next update will be: $\theta_{k+2} = \theta_{k+1} - \frac{\eta_{k+1}}{\tau_{k+1}} \cdot (\gamma^2 v_k + \gamma g_k + g_{k+1})$. The stale gradient $g_k$ is multiplied by $\gamma \frac{\eta_{k+1}}{\tau_{k+1}}$, which is large (assuming $\eta_k \approx \eta_{k+1}$). Despite the fact that $g_k$ was stale, it still has a significant impact on the learning process. In other words, $g_k$ is *under-penalized*.

To eliminate this possibility we penalize the stale gradient itself rather than the learning rate. Using this method, the staleness of each gradient is accounted for within the update vector $v$.

## 5 GAP-AWARE (GA)

In this section we propose a new method called *Gap-Aware* to mitigate over and under-penalization.

### 5.1 THE GAP AS A MEASURE OF GRADIENT STALENESS

An intuitive method to measure gradient staleness would be: $\|\nabla f(\theta_k) - \nabla f(\theta_{k-\tau_k})\|$. This essentially measures the difference between the stale gradient and the accurate gradient that is computed on the up-to-date parameters. (Of course, $\nabla f(\theta_k)$ is never calculated in ASGD algorithms.) Commonly used in deep learning is the Lipschitzian gradients assumption:

$$\|\nabla f(x) - \nabla f(y)\| \leq L\|x - y\|, \quad \forall x, \forall y, L \in \mathbb{R} \tag{2}$$

Setting $x = \theta_k, y = \theta_{k-\tau_k}$ into Equation 2 we get: $\|\nabla f(\theta_k) - \nabla f(\theta_{k-\tau_k})\| \leq L\|\theta_k - \theta_{k-\tau_k}\|$. This implies that $\|\theta_k - \theta_{k-\tau_k}\|$ is a valid (and easily calculated) measure of the gradient staleness. This measure also addresses the delay's over-penalization; using the same simple example described in Section 4.1, the term $\|\theta_k - \theta_{k-\tau_k}\|$ will now be zero, correctly measuring the gradient staleness.

The learning rate $\eta$, commonly decays as the training progresses. This decay can be viewed as a built-in penalization to reduce variance. Following this notion, we suggest reducing the staleness penalization as $\eta$ decays. To accommodate all the attributes above, we define the *Gap*:

**Definition 1.** $G_k$, the Gap at the $k^{th}$ step, is defined as the minimal number of updates required to traverse the current distance between the master's and worker's parameters using the maximal learning rate and assuming all gradients have an average norm. $G_k \in \mathbb{R}$ is defined as:

$$G_k = \|\theta_k - \theta_{k-\tau_k}\|/C + 1$$

Where $C = \eta_{max} \mathbb{E}_k[\|\nabla f(\theta_{k-\tau_k})\|]$ is a constant representing the maximal distance the parameters can travel in a single update, given the gradient's norm is the average gradient norm.

Definition 1 means that dividing $\eta$ by the Gap produces larger steps than those produced by *SA* This allows exploring more distant minimas while still mitigating the gradient staleness. Note that $\mathbb{E}[G_k] = \tau_k$ occurs only if all previous $\tau_k$ updates were in the exact same direction, which rarely happens. Empirically, we found that $\mathbb{E}[G_k] < \tau_k$ (See Appendix C.8).

To mitigate the gradient staleness, while eliminating the over-penalization and under-penalization, we divide the gradients themselves by their respective Gap. We refer to this method as *Gap-Aware* (GA).

## 5.2 CONVERGENCE ANALYSIS

In this section we provide the outlines of the convergence analysis. The complete proofs are given in Appendix B. The GA-ASGD update rule (without momentum) is:

$$\theta_{k+1} = \theta_k - \eta_k \left(\frac{1}{G_k}\right) \cdot \sum_{b=1}^{B} \nabla F(\theta_{k-\tau_k}; \xi_{k-\tau_k,b}) \tag{3}$$

The convergence rate is the speed (or number of steps) at which a convergent sequence approaches its limit. We follow ideas similar to Lian et al. (2015) to show that the upper bound of the convergence rate of GA on a non-convex loss function is similar to that of SGD.

**Assumption 1.** We assume the following, commonly-used assumptions, hold:

- **Unbiased gradient:** The stochastic gradient $\nabla F(\theta; \xi)$ is unbiased:

$$\nabla f(\theta) = \mathbb{E}_\xi[\nabla F(\theta; \xi)] \tag{4}$$

- **Bounded variance:** The variance of the stochastic gradient is bounded:

$$\mathbb{E}_\xi[\|\nabla F(\theta; \xi) - \nabla f(\theta)\|^2] \leq \sigma^2, \quad \forall \theta \tag{5}$$

- **Lipschitzian gradients:** See Equation 2.

- **Independence:** All the random variables $\{\xi_{k,b}\}_{k=1\ldots K; b=1\ldots B}$, are independent.

- **Bounded age:** All delay variables $\tau_1, \ldots \tau_K$ are bounded:

$$\max_k \tau_k \leq T \tag{6}$$

**Theorem 1.** *Assume that Assumption 1 holds and the learning rate sequence $\{\eta_k\}_{k=1\ldots K}$ satisfies:*

$$\frac{BL\eta_k}{G_k} + 2B^2L^2T \sum_{t=1}^{T} \frac{\eta_{k+t}^2}{G_{k+t}^2} \leq 1 \quad \text{for all } k = 1, 2, \ldots \tag{7}$$

*We have the following ergodic convergence rate for the iteration of GA-ASGD:*

$$\frac{1}{\sum_{k=1}^{K} \frac{\eta_k}{G_k}} \sum_{k=1}^{K} \frac{\eta_k}{G_k} \mathbb{E}(\|\nabla f(\theta_k)\|^2) \leq \frac{\frac{2(f(\theta_1)-f(\theta^*))}{B} + \sum_{k=1}^{K} \left(\frac{\eta_k}{G_k} + 2BL \sum_{j=k-T}^{k-1} \frac{\eta_j^2}{G_j^2}\right) \frac{\eta_k L\sigma^2}{G_k}}{\sum_{k=1}^{K} \frac{\eta_k}{G_k}} \tag{8}$$

*Where $\mathbb{E}[\cdot]$ denotes taking expectation in terms of all random variables.*

To simplify the upper bound in Theorem 1, we observed that setting the learning rate $\eta_k$ such that the expression $\frac{\eta_k}{G_k}$ is a constant value across all $k$ obtains the following convergence rate:

**Corollary 1.** *Assume that Assumption 1 holds and that $\eta_k$ are set such that $\frac{\eta_k}{G_k}$ is constant for any $k$ as follows:*

$$\frac{\eta_k}{G_k} := \eta = \sqrt{\frac{f(\theta_1) - f(\theta^*)}{BLK\sigma^2}}, \quad \forall k \in [1, \ldots, K] \tag{9}$$

*If the maximal delay parameter $T$ satisfies:*

$$K \geq \frac{4BL(T+1)^2(f(\theta_1) - f(\theta^*))}{\sigma^2} \tag{10}$$

*then the output of GA satisfies the following ergodic convergence rate:*

$$\min_{k \in \{1, \cdots, K\}} \mathbb{E}[\|\nabla f(\theta_k)\|^2] \leq \frac{1}{K} \sum_{k=1}^{K} \mathbb{E}[\|\nabla f(\theta_k)\|^2] \leq 4\sqrt{\frac{(f(\theta_1) - f(\theta^*))L\sigma^2}{BK}} \tag{11}$$

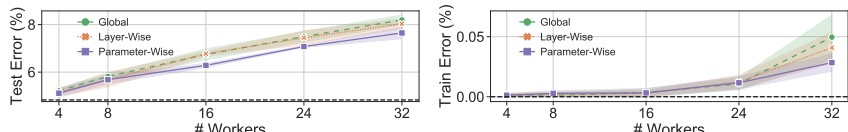

Figure 1: Final test and train error for different numbers of asynchronous workers $N$. The figure shows the average (bold line) and standard deviation (band) of 5 runs on the CIFAR10 dataset using the WideResNet model. The black dashed line is the SGD error using a single worker.

Corollary 1 claims that if the total iterations $K$ is greater than $\mathcal{O}(BT^2)$, the convergence rate achieves $\mathcal{O}(1/\sqrt{BK})$, which is consistent with the convergence rate of ASGD presented in Lian et al. (2015), and with the convergence rate of SGD.

### 5.3 GAP-AWARE VERSIONS

We explore three ways to measure $G_k$:

- **Global:** $G_k \in \mathbb{R}$ as defined in Definition 1.
- **Layer-wise:** Every layer is penalized differently and independently. $G_k \in \mathbb{R}^P$ where $P$ is the number of layers in the model. We denote $\mathbf{1^S}$ as an S-dimensional vector of ones. We denote any vector $X_{*,p}$ is the $p^{th}$ layer in the vector $X_*$. Every element in $G_k$ is calculated per-layer:

$$G_{k,p} = \|\theta_{k,p} - \theta_{k-\tau_k,p}\|/C_p + \mathbf{1^P} \tag{12}$$

- **Parameter-wise:** Every parameter (element in the parameter vector $\theta$) is penalized differently and independently. $G_k \in \mathbb{R}^d$ where $d$ is the number of parameters. We denote $|\cdot|$ on vector $X$, as the absolute value per element of $X$. Every element in $G_k$ is calculated and applied per-element:

$$G_k = |\theta_k - \theta_{k-\tau_k}|/C + \mathbf{1^d} \tag{13}$$

Where $C \in \mathbb{R}^d$ is also calculated element-wise. Specifically, $C = \eta_{max}\mathbb{E}_k[|\nabla f(\theta_{k-\tau_k})|]$.

We tested these variations on three different frameworks[3] to determine which technique has the best performance. Figure 1 demonstrates that the *parameter-wise* method (equation 13) resulted in the best test and train error. Since this phenomenon repeats across all frameworks, we henceforth use the *parameter-wise* method in the Gap-Aware algorithm. We denote $\odot$ as an element-wise multiplication between vectors and describe the final GA algorithm of the master as Algorithm 4. The worker algorithm remains the same as in Algorithm 1.

## 6 EXPERIMENTS

We simulated multiple distributed workers[4] to measure the final test error, train error, and convergence speed of different cluster sizes. To validate that penalizing linearly to the *Gap* is the factor that leads to better performance, we used the same hyperparameters across all the tested algorithms (see Appendix C.4). These hyperparameters are the ones tuned for a single worker, suggested by the authors of the respective papers for each framework. We simulated the workers' execution time using a *gamma-distributed model* (Ali et al., 2000) (see Appendix C.3), where the execution time for each individual batch was drawn from a gamma distribution. The gamma distribution is a well-accepted model for task execution time, which naturally gives rise to stragglers. The importance of asynchronous over synchronous training is explained in Appendix D.

**Combining GA with DANA** One way to verify whether it is better to penalize using the Gap or the delay, is to change the Gap while fixing the delay, and examining the results using GA and SA. Momentum generally increases the norm of the update vector; this in turn, increases the effective step size thus increasing the Gap for a given delay. Hakimi et al. (2019) introduced DANA, which uses the

---

[3]A framework is a unique combination of dataset and model. See full experiment in Appendix C.9.

[4]A worker is a machine with one or more accelerators (i.e., GPU). ASGD methods treat each machine with multiple accelerators, all working synchronously locally, as a single worker.

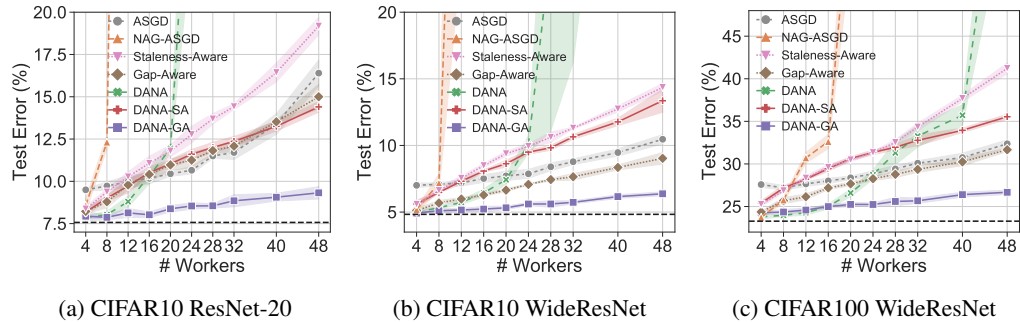

Figure 2: Final test error for different numbers of asynchronous workers $N$. Each line in the figure represents the average (bold line) and standard deviation (band) of 5 runs on a specific framework. The black dashed line represents the average result of SGD using a single worker.

momentum to estimate the master's parameters at the time of the gradient update, thus decreasing the *Gap*. The combination of decreasing the *Gap* using DANA and penalizing stale gradients using GA or SA is easily integrated since both methods are orthogonal. In our experiments, we also compared between DANA-Gap-Aware (DANA-GA) and DANA-Staleness-Aware (DANA-SA). We note that since DANA decreases the Gap, DANA-GA penalizes much less than DANA-SA (see Appendix C.8).

**Algorithms**   Our evaluations consist of the following algorithms:

- *Baseline:* Single worker with the same tuned hyperparameters as in the respective framework's paper. See Appendix C.4.
- *ASGD:* ASGD (Algorithm 2) without momentum ($\gamma = 0$).
- *NAG-ASGD:* ASGD with momentum (Algorithm 2) using a NAG optimizer.
- *Staleness-Aware:* SA as described in Algorithm 3, using a NAG optimizer.
- *Gap-Aware:* Parameter-wise GA (Algorithm 4) as described in Section 5, using a NAG optimizer.
- *DANA:* DANA (Algorithm 6) as described in Appendix C.1.
- *DANA-SA:* DANA-Staleness-Aware (Algorithm 7) as described in Appendix C.1.
- *DANA-GA:* DANA-Gap-Aware (Algorithm 8) as described in Appendix C.1.
- *Adam:* Adam (Algorithm 9) as described in Appendix C.1.
- *Adam-SA:* Adam-Staleness-Aware (Algorithm 10) as described in Appendix C.1.
- *Adam-GA:* Adam-Gap-Aware (Algorithm 11) as described in Appendix C.1.

Our evaluation was extensive on image classification tasks such as CIFAR10, CIFAR100 (Krizhevsky, 2012), and ImageNet (Russakovsky et al., 2015). It also included a language modeling task using the WikiText-103 corpus (Merity et al., 2016). All datasets and models are detailed in Appendix C.2.

### 6.1 EVALUATION ON CIFAR

**Gradient Staleness Effects**   In Figure 2, NAG-ASGD shows how gradient staleness is exacerbated by momentum. NAG-ASGD yields high accuracy with few workers, but the test error climbs sharply when more than 16 workers are used. On the other hand, ASGD without momentum performs poorly using few workers. When using many workers, ASGD significantly surpasses NAG-ASGD because of the implicit momentum generated in asynchronous training (Mitliagkas et al., 2016).

**SA & GA**   Figure 2 also demonstrates that both staleness penalization methods (GA and SA) out-perform the naive NAG-ASGD. GA results in better final test error than SA across all experiments. This empirically proves that GA is the better method for penalizing the gradients. We claim this occurs mainly because SA over-penalizes the gradients, thereby making it impossible to reach any distant, good minima when the number of steps is limited (for more details see Appendix C.8).

**DANA Versions**   Figure 2 shows that DANA potentially diverges when $N$ grows as opposed to DANA-GA and DANA-SA. This shows that DANA benefits from staleness penalization. Furthermore, DANA-GA out-performs all other methods and remains close to the baseline's error across all frameworks. The fact that DANA-GA out-performs DANA-SA validates that GA is superior to SA.

Table 1: Test accuracy on different frameworks[a]. N=32.

| Framework | Tuned ASGD | GA | DANA-GA |
|---|---|---|---|
| **C10 R** | 88% | 87.9% | **91.1%** |
| **C10 WR** | 91.6% | 92.3% | **94.3%** |
| **C100 WR** | 71.1% | 70.6% | **74.3%** |

[a]C10/C100=CIFAR10/100, R=ResNet, WR=WideResNet.

Table 2: Final test perplexity using Transformer-XL on WikiText-103. (Baseline $24.25$. Lower is better).

| N | Adam | Adam-SA | Adam-GA |
|---|---|---|---|
| 4 | 1644.76 | 1210.8 | **26.48** |
| 8 | 1603.01 | 1129.9 | **28.7** |

Table 3: ResNet-50 ImageNet final test accuracy (Baseline $75.64\%$)

| N | ASGD | NAG-ASGD | SA | GA | DANA | DANA-SA | DANA-GA |
|---|---|---|---|---|---|---|---|
| 32 | 70.53% | 70.64% | 61.73% | 70.27% | 74.89% | 65.66% | **75.06%** |
| 48 | 69.05% | 66.78% | 56.22% | 67.75% | 73.75% | 61.16% | **74.23%** |
| 64 | 67.1% | 59.81% | 50.79% | 64.78% | 69.88% | 56.98% | **74.11%** |
| 128 | NaN | NaN | NaN | NaN | NaN | NaN | **72.18%** |

**Tuned ASGD**   To validate that the staleness penalization helps overcome the gradient staleness and improve the results, we tuned the momentum and learning rate of ASGD using 32 workers on the 3 frameworks shown in Figure 2. For each framework, we performed a grid search of 70 perturbations (See Appendix C.12). Table 1 shows that GA and DANA-GA, using the same hyperparameters as the baseline, provide similar or better results than tuning both $\gamma$ and $\eta$, which is highly time-consuming.

According to Mitliagkas et al. (2016), if momentum is used, the asynchronous implicit momentum should impede the convergence as N increases. However, GA and DANA-GA, which use a large momentum, generally perform better than the tuned ASGD even when $N$ is large. This phenomenon repeats across all frameworks, which suggests that GA, and especially DANA-GA, can mitigate the asynchronous implicit momentum problem. Tuning these methods should further improve the results.

The graphs of the train error also show the same concepts discussed here regarding the test error and are presented in Figure 6, Appendix C.10. The convergence rate analysis appears in Appendix C.11.

## 6.2   IMAGENET EXPERIMENTS

We conducted experiments on the ImageNet dataset using the ResNet-50 model (He et al., 2016). Every asynchronous worker is a machine with 8 GPUs, so the 128 workers in our experiments simulate a total of 1024 GPUs. For reference, Goyal et al. (2017) used 256 GPUs synchronously. The hyperparameters we used are those of the tuned single worker (see Appendix C.4). Table 3 shows that GA out-performs SA due to the high number of workers, which exacerbates the over-penalizing of SA. Unlike SA, GA out-performs NAG-ASGD as $N$ increases due to successful staleness mitigation. DANA-GA remains close to the baseline and better than any other method as $N$ increases. DANA-GA reaches 72.18% final test accuracy when using 128 workers, which is the most asynchronous workers shown to converge on ImageNet as far as we know.

## 6.3   NLP EXPERIMENTS

NLP tasks are usually trained using Adam (Kingma & Ba, 2015). To test SA and GA we implemented a version of Adam-SA and Adam-GA given by Algorithm 10 and 11, respectively (Appendix C.1). Transformer-XL (Dai et al., 2019) is a state-of-the-art model for NLP tasks; however, its sensitivity to gradient staleness is catastrophic (Aji & Heafield, 2019). Table 2 shows that GA successfully mitigates the gradient staleness and achieves near-baseline perplexity while SA results in a higher perplexity by almost two orders of magnitude. In this scenario, SA completely fails to mitigate the gradient staleness, proving the superiority of GA. (See hyperparameters in Appendix C.4).

## 7    CONCLUSIONS

The goal of this work is to mitigate gradient staleness, one of the main challenges of ASGD. We argue that penalizing stale gradients linearly to the *delay*, as done in the widely used SA method, flounders due to over and under-penalization. We defined the *Gap* to measure gradient staleness and proposed GA, a novel asynchronous distributed technique that mitigates the gradient staleness by penalizing stale gradients linearly to the Gap. We showed that GA surpasses SA across all frameworks, especially in NLP problems or when the number of workers is large. This presents GA as a superior alternative for staleness penalizing. We further introduced DANA-GA and demonstrated that DANA-GA mitigates gradient staleness better than any of the other methods we compared. Despite prior belief, DANA-GA's superb performance enables the use of momentum in asynchronous environments with many workers; it presents a desirable alternative for parallel training with multiple workers, especially on non-dedicated environments such as cloud computing. In future work, we plan to examine what makes GA perform so well in NLP tasks.

## ACKNOWLEDGEMENT

This work was supported by The Hasso Plattner Institute.

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

## A  SOURCE CODE

The source code of DANA-Gap-Aware is provided via:
DOWNLOAD LINK

## B  PROOFS

**Proof for Theorem 1**

*Proof.* From the Lipschitzisan gradient assumption equation 2, we have

$$
\begin{aligned}
f(\theta_{k+1}) - f(\theta_k) \leq \\
\leq \langle \nabla f(\theta_k), \theta_{k+1} - \theta_k \rangle + \frac{L}{2} \|\theta_{k+1} - \theta_k\|^2 = \\
= - \left\langle \nabla f(\theta_k), \eta_k \left(\frac{1}{G_k}\right) \cdot \sum_{b=1}^{B} \nabla F(\theta_{k-\tau_k}; \xi_{k-\tau_k,b}) \right\rangle + \frac{\eta_k^2 L}{2G_k^2} \left\| \sum_{b=1}^{B} \nabla F(\theta_{k-\tau_k}; \xi_{k-\tau_k,b}) \right\|^2 \\
= - \frac{B\eta_k}{G_k} \left\langle \nabla f(\theta_k), \frac{1}{B} \sum_{b=1}^{B} \nabla F(\theta_{k-\tau_k}; \xi_{k-\tau_k,b}) \right\rangle + \frac{\eta_k^2 L}{2G_k^2} \left\| \sum_{b=1}^{B} \nabla F(\theta_{k-\tau_k}; \xi_{k-\tau_k,b}) \right\|^2
\end{aligned}
\tag{14}
$$

Taking expectation with respect to $\xi_{k,*}$ on both sides of equation 14, we have

$$
\begin{aligned}
\mathbb{E}_{\xi_{k,*}}[f(\theta_{k+1})] - f(\theta_k) \leq & - \frac{B\eta_k}{G_k} \left\langle \nabla f(\theta_k), \frac{1}{B} \sum_{b=1}^{B} \nabla f(\theta_{k-\tau_k}) \right\rangle \\
& + \frac{\eta_k^2 L}{2G_k^2} \mathbb{E}_{\xi_{k,*}} \left[ \left\| \sum_{b=1}^{B} \nabla F(\theta_{k-\tau_k}; \xi_{k-\tau_k,b}) \right\|^2 \right]
\end{aligned}
\tag{15}
$$

where we use the unbiased stochastic gradient assumption (equation 4).

From the fact that:

$$
\langle a, b \rangle = \frac{1}{2} \left( \|a\|^2 + \|b\|^2 - \|a-b\|^2 \right)
$$

we have

$$
\begin{aligned}
& \mathbb{E}_{\xi_{k,*}}[f(\theta_{k+1})] - f(\theta_k) \\
& \leq - \frac{B\eta_k}{2G_k} \left( \|\nabla f(\theta_k)\|^2 + \left\| \frac{1}{B} \sum_{b=1}^{B} \nabla f(\theta_{k-\tau_k}) \right\|^2 - \underbrace{\left\| \nabla f(\theta_k) - \frac{1}{B} \sum_{b=1}^{B} \nabla f(\theta_{k-\tau_k}) \right\|^2}_{T_1} \right) \\
& \quad + \frac{\eta_k^2 L}{2G_k^2} \underbrace{\mathbb{E}_{\xi_{k,*}} \left[ \left\| \sum_{b=1}^{B} \nabla F(\theta_{k-\tau_k}; \xi_{k-\tau_k,b}) \right\|^2 \right]}_{T_2}
\end{aligned}
\tag{16}
$$

Next we estimate the upper bound of $T_1$ and $T_2$. For $T_2$ we have

$$
\begin{aligned}
T_2 =& \mathbb{E}_{\xi_{k,*}} \left[ \left\| \sum_{b=1}^{B} \nabla F(\theta_{k-\tau_k}; \xi_{k-\tau_k,b}) \right\|^2 \right] \\
=& \mathbb{E}_{\xi_{k,*}} \left[ \left\| \left( \sum_{b=1}^{B} (\nabla F(\theta_{k-\tau_k}; \xi_{k-\tau_k,b}) - \nabla f(\theta_{k-\tau_k})) + \sum_{b=1}^{B} \nabla f(\theta_{k-\tau_k}) \right) \right\|^2 \right] \\
=& \mathbb{E}_{\xi_{k,*}} \left[ \left\| \sum_{b=1}^{B} (\nabla F(\theta_{k-\tau_k}; \xi_{k-\tau_k,b}) - \nabla f(\theta_{k-\tau_k})) \right\|^2 + \left\| \sum_{b=1}^{B} \nabla f(\theta_{k-\tau_k}) \right\|^2 \right. \\
& \left. + 2 \left\langle \sum_{b=1}^{B} (\nabla F(\theta_{k-\tau_k}; \xi_{k-\tau_k,b}) - \nabla f(\theta_{k-\tau_k})), \sum_{b=1}^{B} \nabla f(\theta_{k-\tau_k}) \right\rangle \right] \\
=& \mathbb{E}_{\xi_{k,*}} \left[ \left\| \sum_{b=1}^{B} (\nabla F(\theta_{k-\tau_k}; \xi_{k-\tau_k,b}) - \nabla f(\theta_{k-\tau_k})) \right\|^2 + \left\| \sum_{b=1}^{B} \nabla f(\theta_{k-\tau_k}) \right\|^2 \right] \\
=& \mathbb{E}_{\xi_{k,*}} \left[ \left\| \sum_{b=1}^{B} (\nabla F(\theta_{k-\tau_k}; \xi_{k-\tau_k,b}) - \nabla f(\theta_{k-\tau_k})) \right\|^2 + \left\| \sum_{b=1}^{B} \nabla f(\theta_{k-\tau_k}) \right\|^2 \right] \\
=& \mathbb{E}_{\xi_{k,*}} \left[ \sum_{b=1}^{B} \|(\nabla F(\theta_{k-\tau_k}; \xi_{k-\tau_k,b}) - \nabla f(\theta_{k-\tau_k}))\|^2 \right. \\
& + 2 \sum_{1 \le b < b' \le B} \langle \nabla F(\theta_{k-\tau_k}; \xi_{k-\tau_k,b}) - \nabla f(\theta_{k-\tau_k}), \nabla F(\theta_{k-\tau_k}; \xi_{k-\tau_k,b'}) - \nabla f(\theta_{k-\tau_k}) \rangle \\
& \left. + \left\| \sum_{b=1}^{B} \nabla f(\theta_{k-\tau_k}) \right\|^2 \right] \\
\le& \mathbb{E}_{\xi_{k,*}} \left[ \sum_{b=1}^{B} \left( \|(\nabla F(\theta_{k-\tau_k}; \xi_{k-\tau_k,b}) - \nabla f(\theta_{k-\tau_k}))\|^2 \right) + \left\| \sum_{b=1}^{B} \nabla f(\theta_{k-\tau_k}) \right\|^2 \right] \\
\le& B\sigma^2 + \left\| \sum_{b=1}^{B} \nabla f(\theta_{k-\tau_k}) \right\|^2
\end{aligned}
\tag{17}
$$

where the fifth equality is due to

$$
\begin{aligned}
& \mathbb{E}_{\xi_{k,*}} \left\langle \sum_{b=1}^{B} (\nabla F(\theta_{k-\tau_k}; \xi_{k-\tau_k,b}) - \nabla f(\theta_{k-\tau_k})), \sum_{b=1}^{B} \nabla f(\theta_{k-\tau_k}) \right\rangle \\
=& \left\langle \sum_{b=1}^{B} \mathbb{E}_{\xi_{k,*}} (\nabla F(\theta_{k-\tau_k}; \xi_{k-\tau_k,b}) - \nabla f(\theta_{k-\tau_k})), \sum_{b=1}^{B} \nabla f(\theta_{k-\tau_k}) \right\rangle \\
=& 0
\end{aligned}
$$

and the last inequality is due to the bounded variance assumption equation 5 and due to:

$$
\begin{aligned}
&\mathbb{E}_{\xi_{k,*}}\left[\sum_{1\le b<b'\le B}\langle\nabla F(\theta_{k-\tau_k};\xi_{k-\tau_k,b})-\nabla f(\theta_{k-\tau_k}),\nabla F(\theta_{k-\tau_k};\xi_{k-\tau_k,b'})-\nabla f(\theta_{k-\tau_k})\rangle\right]\\
=&\mathbb{E}_{\xi_{k,*}}\left[\sum_{1\le b<b'\le B}\mathbb{E}_{k-\tau_k,b'}\left[\langle\nabla F(\theta_{k-\tau_k};\xi_{k-\tau_k,b})-\nabla f(\theta_{k-\tau_k}),\nabla F(\theta_{k-\tau_k};\xi_{k-\tau_k,b'})-\nabla f(\theta_{k-\tau_k})\right]\right]\\
=&\mathbb{E}_{\xi_{k,*}}\left[\sum_{1\le b<b'\le B}\langle\nabla F(\theta_{k-\tau_k};\xi_{k-\tau_k,b})-\nabla f(\theta_{k-\tau_k}),\mathbb{E}_{k-\tau_k,b'}\left[\nabla F(\theta_{k-\tau_k};\xi_{k-\tau_k,b'})-\nabla f(\theta_{k-\tau_k})\right]\rangle\right]\\
=&0.
\end{aligned}
\tag{18}
$$

We next turn to $T_1$:

$$
\begin{aligned}
T_1&=\left\|\nabla f(\theta_k)-\frac{1}{B}\sum_{b=1}^{B}\nabla f\left(\theta_{k-\tau_k}\right)\right\|^2\\
&=\left\|\nabla f(\theta_k)-\nabla f\left(\theta_{k-\tau_k}\right)\right\|^2\\
&\le L^2\|\theta_k-\theta_{k-\tau_k}\|^2
\end{aligned}
$$

where the last inequality is from the Lipschitzian gradient assumption (equation 2). It follows that

$$
\begin{aligned}
T_1&\le L^2\left\|\theta_k-\theta_{k-\tau_k}\right\|^2\\
&=L^2\left\|\sum_{j=k-\tau_k}^{k-1}(\theta_{j+1}-\theta_j)\right\|^2\\
&=L^2\left\|\sum_{j=k-\tau_k}^{k-1}\eta_j\left(\frac{1}{G_j}\right)\cdot\sum_{b=1}^{B}\nabla F\left(\theta_{j-\tau_j};\xi_{j-\tau_j,b}\right)\right\|^2\\
&=L^2\left\|\sum_{j=k-\tau_k}^{k-1}\eta_j\left(\frac{1}{G_j}\right)\cdot\sum_{b=1}^{B}\left[\nabla F\left(\theta_{j-\tau_j};\xi_{j-\tau_j,b}\right)-\nabla f\left(\theta_{j-\tau_j}\right)\right]+\sum_{j=k-\tau_k}^{k-1}\eta_j\left(\frac{1}{G_j}\right)\cdot\sum_{b=1}^{B}\nabla f\left(\theta_{j-\tau_j}\right)\right\|^2\\
&\le 2L^2\left(\underbrace{\left\|\sum_{j=k-\tau_k}^{k-1}\eta_j\left(\frac{1}{G_j}\right)\cdot\sum_{b=1}^{B}\left[\nabla F\left(\theta_{j-\tau_j};\xi_{j-\tau_j,b}\right)-\nabla f\left(\theta_{j-\tau_j}\right)\right]\right\|^2}_{T_3}+\underbrace{\left\|\sum_{j=k-\tau_k}^{k-1}\eta_j\left(\frac{1}{G_j}\right)\cdot\sum_{b=1}^{B}\nabla f\left(\theta_{j-\tau_j}\right)\right\|^2}_{T_4}\right)
\end{aligned}
\tag{19}
$$

where the last inequality uses the fact that $\|a + b\|^2 \leq 2\|a\|^2 + 2\|b\|^2$ for any real vectors $a$ and $b$. Taking the expectation in terms of $\{\xi_{j-\tau_j,*}|j \in \{k - \tau_k, ..., k - 1\}\}$ for $T_3$, we have

$$
\mathbb{E}_{\xi_{j-\tau_j,*}|j\in\{k-\tau_k,...,k-1\}}(T_3)
$$

$$
=\mathbb{E}_{\xi_{j-\tau_j,*}|j\in\{k-\tau_k,...,k-1\}}\left[\left\|\sum_{j=k-\tau_k}^{k-1}\eta_j\left(\frac{1}{G_j}\right)\cdot\sum_{b=1}^{B}\left(\nabla F(\theta_{j-\tau_j};\xi_{j-\tau_j,b})-\nabla f(\theta_{j-\tau_j})\right)\right\|^2\right]
$$

$$
=\mathbb{E}_{\xi_{j-\tau_j,*}|j\in\{k-\tau_k,...,k-1\}}\left[\sum_{j=k-\tau_k}^{k-1}\frac{\eta_j^2}{G_j^2}\left\|\sum_{b=1}^{B}\left(\nabla F(\theta_{j-\tau_j};\xi_{j-\tau_j,b})-\nabla f(\theta_{j-\tau_j})\right)\right\|^2\right]
$$

$$
+2\mathbb{E}_{\xi_{j-\tau_j,*}|j\in\{k-\tau_k,...,k-1\}}\left[\sum_{k-1\geq j''>j'\geq k-\tau_k}\frac{\eta_{j'}\eta_{j''}}{G_j'G_j''}\left\langle\sum_{b=1}^{B}\left(\nabla F(\theta_{j''-\tau_{j''}};\xi_{j''-\tau_{j''},b})-\nabla f(\theta_{j''-\tau_{j''}})\right),\right.\right.
$$

$$
\left.\left.\sum_{b=1}^{B}\left(\nabla F(\theta_{j'-\tau_{j'}};\xi_{j'-\tau_{j'},b})-\nabla f(\theta_{j'-\tau_{j'}})\right)\right\rangle\right]
$$

$$
=\mathbb{E}_{\xi_{j-\tau_j,*}|j\in\{k-\tau_k,...,k-1\}}\left[\sum_{j=k-\tau_k}^{k-1}\frac{\eta_j^2}{G_j^2}\left\|\sum_{b=1}^{B}\left(\nabla F(\theta_{j-\tau_j};\xi_{j-\tau_j,b})-\nabla f(\theta_{j-\tau_j})\right)\right\|^2\right]
$$

$$
=\mathbb{E}_{\xi_{j-\tau_j,*}|j\in\{k-\tau_k,...,k-1\}}\left[\sum_{j=k-\tau_k}^{k-1}\frac{\eta_j^2}{G_j^2}\sum_{b=1}^{B}\left\|\left(\nabla F(\theta_{j-\tau_j};\xi_{j-\tau_j,b})-\nabla f(\theta_{j-\tau_j})\right)\right\|^2\right]
$$

$$
\leq\mathbb{E}_{\xi_{j-\tau_j,*}|j\in\{k-\tau_k,...,k-1\}}\left[\sum_{j=k-\tau_k}^{k-1}\frac{B\eta_j^2}{G_j^2}\left\|\left(\nabla F(\theta_{j-\tau_j};\xi_{j-\tau_j,b})-\nabla f(\theta_{j-\tau_j})\right)\right\|^2\right]
$$

$$
\leq B\sum_{j=k-\tau_k}^{k-1}\frac{\eta_j^2}{G_j^2}\sigma^2 \tag{20}
$$

where the second to last equality is due to the last lines in equation 17 and the third equality is due to

$$
\mathbb{E}_{\xi_j,k-1\geq j\geq k-\tau_k}\left[\sum_{k+\tau_k-1\geq j''>j'\geq k}\frac{\eta_{j'}\eta_{j''}}{G_j'G_j''}\left\langle\sum_{b=1}^{B}\left(\nabla F(\theta_{j''-\tau_{j''}};\xi_{j''-\tau_{j''},b})-\nabla f(\theta_{j''-\tau_{j''}})\right),\right.\right.
$$

$$
\left.\left.\sum_{b=1}^{B}\left(\nabla F(\theta_{j'-\tau_{j'}};\xi_{j'-\tau_{j'},b})-\nabla f(\theta_{j'-\tau_{j'}})\right)\right\rangle\right]
$$

$$
=\mathbb{E}_{\xi_j,k-1\geq j\geq k-\tau_k}\left[\sum_{k-1\geq j''>j'\geq k-\tau_k}\frac{\eta_{j'}\eta_{j''}}{G_j'G_j''}\left\langle\sum_{b=1}^{B}\mathbb{E}_{j'',*}\left[\nabla F(\theta_{j''-\tau_{j''}};\xi_{j''-\tau_{j''},b})-\nabla f(\theta_{j''-\tau_{j''}})\right],\right.\right.
$$

$$
\left.\left.\sum_{b=1}^{B}\left(\nabla F(\theta_{j'-\tau_{j'}};\xi_{j'-\tau_{j'},b})-\nabla f(\theta_{j'-\tau_{j'}})\right)\right\rangle\right]
$$

$$
=0.
$$

Taking the expectation in terms of $\xi_{j-\tau_j,*}$ for $T_4$, we have

$$
\mathbb{E}_{\xi_{j-\tau_j,*}|j\in\{k-\tau_k,...,k-1\}}[T_4]
$$

$$
=\mathbb{E}_{\xi_{j-\tau_j,*}|j\in\{k-\tau_k,...,k-1\}}\left[\left\|\sum_{j=k-\tau_k}^{k-1}\eta_j\left(\frac{1}{G_j}\right)\cdot\sum_{b=1}^{B}\nabla f\left(\theta_{j-\tau_j}\right)\right\|^2\right]
$$

$$
=\mathbb{E}_{\xi_{j-\tau_j,*}|j\in\{k-\tau_k,...,k-1\}}\left[\left\|\sum_{j=k-\tau_k}^{k-1}\frac{B\eta_j}{G_j}\cdot\nabla f\left(\theta_{j-\tau_j}\right)\right\|^2\right]
$$

$$
\leq T\sum_{j=k-\tau_k}^{k-1}\frac{B^2\eta_j^2}{G_j^2}\mathbb{E}_{\xi_{j-\tau_j,*}|j\in\{k-\tau_k,...,k-1\}}\left[\left\|\nabla f(\theta_{j-\tau_j})\right\|^2\right] \tag{21}
$$

where the last inequality uses the fact that $\|\sum_{i=1}^{N}a_i\|^2\leq N\sum_{i=1}^{N}\|a_i\|^2$ for any real vectors $a_i$ and the bounded age assumption equation 6.

We take full expectation on both sides of equation 19 and substitute $\mathbb{E}[T_3]$ and $\mathbb{E}[T_4]$ by their upper bounds, equation 20 and equation 21 respectively:

$$
\mathbb{E}[T_1]\leq 2L^2\left(B\sum_{j=k-\tau_k}^{k-1}\frac{\eta_j^2}{G_j^2}\sigma^2+T\sum_{j=k-\tau_k}^{k-1}\frac{B^2\eta_j^2}{G_j^2}\mathbb{E}\left[\left\|\nabla f(\theta_{j-\tau_j})\right\|^2\right]\right) \tag{22}
$$

Substituting $\mathbb{E}[T_1]$ and $\mathbb{E}[T_2]$ with their upper bounds equation 22 and equation 17 respectively, and taking full expectation on both sides in equation 16, we obtain

$$
\mathbb{E}[f(\theta_{k+1})]-f(\theta_k)
$$

$$
\leq-\frac{B\eta_k}{2G_k}\left(\mathbb{E}[\|\nabla f(\theta_k)\|^2]+\mathbb{E}\left[\left\|\frac{1}{B}\sum_{b=1}^{B}\nabla f(\theta_{k-\tau_k})\right\|^2\right]\right)
$$

$$
+\frac{B\eta_k L^2}{G_k}\left(B\sum_{j=k-\tau_k}^{k-1}\frac{\eta_j^2}{G_j^2}\sigma^2+T\sum_{j=k-\tau_k}^{k-1}\frac{B^2\eta_j^2}{G_j^2}\mathbb{E}\left[\left\|\nabla f(\theta_{j-\tau_j})\right\|^2\right]\right)
$$

$$
+\frac{\eta_k^2 L}{2G_k^2}\left(B\sigma^2+\mathbb{E}\left[\left\|\sum_{b=1}^{B}\nabla f(\theta_{k-\tau_k})\right\|^2\right]\right)
$$

$$
\leq-\frac{B\eta_k}{2G_k}\mathbb{E}[\|\nabla f(\theta_k)\|^2]+\left(\frac{B^2\eta_k^2 L}{2G_k^2}-\frac{B\eta_k}{2G_k}\right)\mathbb{E}\left[\left\|\nabla f(\theta_{k-\tau_k})\right\|^2\right]
$$

$$
+\left(\frac{B\eta_k^2 L}{2G_k^2}+\frac{B^2\eta_k L^2}{G_k}\sum_{j=k-\tau_k}^{k-1}\frac{\eta_j^2}{G_j^2}\right)\sigma^2+\frac{B^3\eta_k L^2 T}{G_k}\sum_{j=k-\tau_k}^{k-1}\frac{\eta_j^2}{G_j^2}\mathbb{E}\left[\left\|\nabla f(\theta_{j-\tau_j})\right\|^2\right] \tag{23}
$$

Summarizing the inequality equation 23 from $k = 1$ to $k = K$, meaning until the K-th update of the master, we have

$$\mathbb{E}[f(\theta_{K+1})] - f(\theta_1)$$

$$\leq -\frac{B}{2}\sum_{k=1}^{K}\frac{\eta_k}{G_k}\mathbb{E}\left[\|\nabla f(\theta_k)\|^2\right] + \sum_{k=1}^{K}\left(\frac{B^2\eta_k^2 L}{2G_k^2} - \frac{B\eta_k}{2G_k}\right)\mathbb{E}\left[\|\nabla f(\theta_{k-\tau_k})\|^2\right]$$

$$+ \sum_{k=1}^{K}\left(\frac{B\eta_k^2 L}{2G_k^2} + \frac{B^2\eta_k L^2}{G_k}\sum_{j=k-\tau_k}^{k-1}\frac{\eta_j^2}{G_j^2}\right)\sigma^2$$

$$+ TL^2\sum_{k=1}^{K}\left(\frac{B^3\eta_k}{G_k}\sum_{j=k-\tau_k}^{k-1}\frac{\eta_j^2}{G_j^2}\mathbb{E}\left[\|\nabla f(\theta_{j-\tau_j})\|^2\right]\right)$$

$$\leq -\frac{B}{2}\sum_{k=1}^{K}\frac{\eta_k}{G_k}\mathbb{E}\left(\|\nabla f(\theta_k)\|^2\right)$$

$$+ \sum_{k=1}^{K}\left(\frac{B^2\eta_k^2 L}{2G_k^2} + \frac{B^3\eta_k L^2 T}{G_k}\sum_{t=1}^{T}\frac{\eta_{k+t}^2}{G_{k+t}^2} - \frac{B\eta_k}{2G_k}\right)\mathbb{E}\left[\|\nabla f(\theta_{k-\tau_k})\|^2\right]$$

$$+ \sum_{k=1}^{K}\left(\frac{B\eta_k^2 L}{2G_k^2} + \frac{B^2\eta_k L^2}{G_k}\sum_{j=k-\tau_k}^{k-1}\frac{\eta_j^2}{G_j^2}\right)\sigma^2$$

$$\leq -\frac{B}{2}\sum_{k=1}^{K}\frac{\eta_k}{G_k}\mathbb{E}\left(\|\nabla f(\theta_k)\|^2\right) + \sum_{k=1}^{K}\left(\frac{B\eta_k^2 L}{2G_k^2} + \frac{B^2\eta_k L^2}{G_k}\sum_{j=k-\tau_k}^{k-1}\frac{\eta_j^2}{G_j^2}\right)\sigma^2$$

$$\leq -\frac{B}{2}\sum_{k=1}^{K}\frac{\eta_k}{G_k}\mathbb{E}\left(\|\nabla f(\theta_k)\|^2\right) + \sum_{k=1}^{K}\left(\frac{B\eta_k^2 L}{2G_k^2} + \frac{B^2\eta_k L^2}{G_k}\sum_{j=k-T}^{k-1}\frac{\eta_j^2}{G_j^2}\right)\sigma^2 \qquad (24)$$

where the second to last inequality is due to equation 7 and the last inequality is due to equation 6. Note that $\theta^*$ is the global optimization point. By doing a few simple algebraic operations on equation 24 we have:

$$\frac{1}{\sum_{k=1}^{K}\frac{\eta_k}{G_k}}\sum_{k=1}^{K}\frac{\eta_k}{G_k}\mathbb{E}(\|\nabla f(\theta_k)\|^2) \leq \frac{2(f(\theta_1) - f(\theta^*)) + \sum_{k=1}^{K}\left(\frac{B\eta_k^2 L}{G_k^2} + \frac{2B^2\eta_k L^2}{G_k}\sum_{j=k-\tau_k}^{k+T-1}\frac{\eta_j^2}{G_j^2}\right)\sigma^2}{B\sum_{k=1}^{K}\frac{\eta_k}{G_k}}$$

$$(25)$$

This completes the proof. $\qquad\square$

**Proof for Corollary 1**

*Proof.* Combining equation 9 and equation 10, we get

$$K \geq \frac{4BL(T+1)^2(f(\theta_1) - f(\theta^*))}{\sigma^2}$$

$$\frac{1}{4B^2 L^2(T+1)^2} \geq \frac{(f(\theta_1) - f(\theta^*))}{BLK\sigma^2}$$

$$\frac{1}{4B^2 L^2(T+1)^2} \geq \eta^2$$

$$\eta \leq \frac{1}{2BL(T+1)} \qquad (26)$$

If follows from equation 26 that

$$B\eta L + 2B^2 L^2 T^2\eta^2 \leq \frac{1}{2T+2} + \frac{T^2}{2(T+1)^2} \leq \frac{1}{2} + \frac{1}{2} = 1$$

The last inequality holds since naturally $T \geq 0$. This implies that condition equation 7 in Theorem 1 is satisfied globally. Then we can safely apply equation 8 in Theorem 1:

$$
\begin{aligned}
\frac{1}{K} \sum_{i=1}^{K} \mathbb{E}(\|\nabla f(\theta_i)\|^2) \leq & \frac{2(f(\theta_1) - f(\theta^*)) + K\left(B\eta^2 L + 2B^2 L^2 T\eta^3\right)\sigma^2}{BK\eta} \\
= & \frac{2(f(\theta_1) - f(\theta^*))}{BK\eta} + L\sigma^2\eta\left(1 + 2BLT\eta\right) \\
\leq & \frac{2(f(\theta_1) - f(\theta^*))}{BK\eta} + 2L\sigma^2\eta \\
= & 2\sqrt{\frac{(f(\theta_1) - f(\theta^*))L\sigma^2}{BK}} + 2\sqrt{\frac{(f(\theta_1) - f(\theta^*))L\sigma^2}{BK}} \\
= & 4\sqrt{\frac{(f(\theta_1) - f(\theta^*))L\sigma^2}{BK}}
\end{aligned}
$$

where the third inequality is due to equation 26 and the second to last equality uses equation 9. This completes the proof. □

**Proof for equation in Section 4.1**

*Proof.*

$$
\begin{aligned}
\theta_{k+2} &= \theta_{k+1} - \frac{\eta_{k+1}}{\tau_{k+1}} \cdot v_{k+2} \\
&= \theta_{k+1} - \frac{\eta_{k+1}}{\tau_{k+1}} \cdot (\gamma v_{k+1} + g_{k+1}) \\
&= \theta_{k+1} - \frac{\eta_{k+1}}{\tau_{k+1}} \cdot (\gamma(\gamma v_k + g_k) + g_{k+1}) \\
&= \theta_{k+1} - \frac{\eta_{k+1}}{\tau_{k+1}} \cdot (\gamma^2 v_k + \gamma g_k + g_{k+1})
\end{aligned}
\tag{27}
$$

□

## C  EXPERIMENTAL SETUP

### C.1  ALGORITHMS

Algorithms 5 to 11 only change the master's algorithm; the complementary worker algorithm is the same as ASGD (Algorithm 1). The master's scheme is a simple FIFO. We consider parameter server optimizations beyond the scope of this paper.

The calculation of the $C$ coefficient is described in Appendix C.6.

We note that in Algorithms 8 and 11 the term $\theta^i$ is identical to $\theta_{k-\tau_k}$ from the definition of $\tau_k$ in Section 4.

---

**Algorithm 5** Staleness-Aware-Gradient: master

---

Initialize an iteration array: $iter = [0] * N$
For k = 1...K do:
    Receive gradient $g_k^i$ from worker $i$
    Calculate worker $i$'s current delay $\tau_k \leftarrow k - iter[i]$
    Update momentum $v_{k+1} \leftarrow \gamma v_k + \frac{g_k^i}{\tau_k}$
    Update master's weights $\theta_{k+1} \leftarrow \theta_k - \eta_k \cdot v_{k+1}$
    Send $\theta_{k+1}$ to worker $i$
    Save current iteration $iter[i] \leftarrow k$

---

---

**Algorithm 6** DANA: master

---

For k = 1...K do:
    Receive gradient $g_k^i$ from worker $i$
    Update worker's momentum $v^i \leftarrow \gamma v^i + g_k^i$
    Update master's weights $\theta_{k+1} \leftarrow \theta_k - \eta_k v^i$
    Send estimate $\hat{\theta} = \theta_{k+1} - \eta_k \gamma \sum_{j=1}^{N} v^j$ to worker $i$

---

**Algorithm 7** DANA-Staleness-Aware: master

---

Initialize an iteration array for the workers: $iter = [0] * N$
For k = 1...K do:
    Receive gradient $g_k^i$ from worker $i$
    Calculate worker $i$'s current delay $\tau_k \leftarrow k - iter[i]$
    Update worker's momentum $v^i \leftarrow \gamma v^i + \frac{g_k^i}{\tau_k}$
    Update master's weights $\theta_{k+1} \leftarrow \theta_k - \eta_k v^i$
    Send estimate $\hat{\theta} = \theta_{k+1} - \eta_k \gamma \sum_{j=1}^{N} v^j$ to worker $i$
    Save current iteration $iter[i] \leftarrow k$

---

**Algorithm 8** DANA-Gap-Aware: master

---

Initialize the given weights for each worker: $\theta^i = \theta_0$
For k = 1...K do:
    Receive gradient $g_k^i$ from worker $i$
    Calculate Gap: $G_k = \frac{|\theta_k - \theta^i|}{C} + \mathbf{1}^d$
    Update worker's momentum $v^i \leftarrow \gamma v^i + \left(\frac{1}{G_k}\right) \odot g_k^i$
    Update master's weights $\theta_{k+1} \leftarrow \theta_k - \eta v^i$
    Save and send estimate $\theta^i \leftarrow \theta_{k+1} - \eta \gamma \sum_{j=1}^{N} v^j$ to worker $i$

---

The *Adam*-based algorithm requires a slightly more intelligent integration to the gradient staleness penalizing methods (such as GA and SA). Penalizing the gradient before calculating the first and second moments doesn't affect the update vector since the first and second moments cancel each other out. Therefore, we suggest applying the penalty only on the first moment, thus decreasing the update step's size by the desired amount. Since DANA's integration into the Adam algorithm is convoluted and not straight forward, we chose not to implement the combination in this paper.

---

**Algorithm 9** Adam: master

---

**Require:** $\eta_1 \ldots \eta_K$: step lengths
**Require:** $\beta_1, \beta_2 \in [0, 1)$: exponential decay rates for the moment estimates
**Initialize:** $m_0 \leftarrow 0, \quad v_0 \leftarrow 0$
For k = 1...K do:
    Receive gradient $g_k^i$ from worker $i$
    Update biased first moment estimate $m_k \leftarrow \beta_1 m_{k-1} + (1 - \beta_1) g_k^i$
    Update biased second moment estimate $v_k \leftarrow \beta_2 v_{k-1} + (1 - \beta_2)(g_k^i)^2$
    Compute bias-corrected first order moment estimate $\hat{m}_k \leftarrow \frac{m_k}{1-\beta_1^k}$
    Compute bias-corrected second order moment estimate $\hat{v}_k \leftarrow \frac{v_k}{1-\beta_2^k}$
    Update master's weights $\theta_{k+1} \leftarrow \theta_k - \frac{\eta_k \cdot \hat{m}_k}{\sqrt{\hat{v}_k} + \epsilon}$
    Send $\theta_{k+1}$ to worker $i$

---

---

**Algorithm 10** Adam-Staleness-Aware: master

---

**Require:** $\eta_1 \ldots \eta_K$: step lengths
**Require:** $\beta_1, \beta_2 \in [0, 1)$: exponential decay rates for the moment estimates
**Initialize:** $m_0 \leftarrow 0, \quad v_0 \leftarrow 0$
**Initialize:** $iter = [0] * N$: an iteration array for the workers
For k = 1...K do:
    Receive gradient $g_k^i$ from worker $i$
    Calculate worker $i$'s current delay $\tau_k \leftarrow k - iter[i]$
    Update biased first moment estimate $m_k \leftarrow \beta_1 m_{k-1} + (1 - \beta_1)\frac{g_k^i}{\tau_k}$
    Update biased second moment estimate $v_k \leftarrow \beta_2 v_{k-1} + (1 - \beta_2)(g_k^i)^2$
    Compute bias-corrected first order moment estimate $\hat{m}_k \leftarrow \frac{m_k}{1-\beta_1^k}$
    Compute bias-corrected second order moment estimate $\hat{v}_k \leftarrow \frac{v_k}{1-\beta_2^k}$
    Update master's weights $\theta_{k+1} \leftarrow \theta_k - \frac{\eta_k \cdot \hat{m}_k}{\sqrt{\hat{v}_k}+\epsilon}$
    Send $\theta_{k+1}$ to worker $i$
    Save current iteration $iter[i] \leftarrow k$

---

**Algorithm 11** Adam-Gap-Aware: master

---

**Require:** $\eta_1 \ldots \eta_K$: step lengths
**Require:** $\beta_1, \beta_2 \in [0, 1)$: exponential decay rates for the moment estimates
**Initialize:** $m_0 \leftarrow 0, \quad v_0 \leftarrow 0$
**Initialize:** $\theta^i = \theta_0$: parameters for every worker
For k = 1...K do:
    Receive gradient $g_k^i$ from worker $i$

    Calculate Gap: $G_k = \frac{|\theta_k - \theta^i|}{C} + \mathbf{1}^d$
    Update biased first moment estimate $m_k \leftarrow \beta_1 m_{k-1} + \left(\frac{1-\beta_1}{G_k}\right) \odot g_k^i$
    Update biased second moment estimate $v_k \leftarrow \beta_2 v_{k-1} + (1 - \beta_2)(g_k^i)^2$
    Compute bias-corrected first order moment estimate $\hat{m}_k \leftarrow \frac{m_k}{1-\beta_1^k}$
    Compute bias-corrected second order moment estimate $\hat{v}_k \leftarrow \frac{v_k}{1-\beta_2^k}$
    Update master's weights $\theta_{k+1} \leftarrow \theta_k - \frac{\eta_k \cdot \hat{m}_k}{\sqrt{\hat{v}_k}+\epsilon}$
    Send $\theta_{k+1}$ to worker $i$
    Save worker $i$'s given parameters $\theta^i \leftarrow \theta_{k+1}$

---

## C.2 DATASETS & MODELS

**CIFAR** The CIFAR-10 (Krizhevsky, 2012) dataset comprises 60K RGB images partitioned into 50K training images and 10K test images. Each image contains 32x32 RGB pixels and belongs to 1 of 10 equal-sized classes. CIFAR-100 is similar but has 100 classes. Link.

**ImageNet** The ImageNet dataset (Russakovsky et al., 2015), known as ILSVRC2012, consists of RGB images, each labeled as one of 1000 classes. The images are partitioned into 1.28 million training images and 50K validation images; each image is randomly cropped and re-sized to 224x224 (1-crop validation). Link.

**WikiText-103** The WikiText language modeling dataset is a collection of over 100 million tokens extracted from the set of verified *Good* and *Featured* articles on Wikipedia. Compared to the preprocessed version of Penn Treebank (PTB), WikiText-103 is over 110 times larger. The WikiText dataset also features a far larger vocabulary and retains the original case, punctuation, and numbers (Merity et al., 2016). Link.

**Transformer-XL**   The *WikiText-103* dataset is trained on the *Transformer-XL* model (Dai et al., 2019). The hyperparameters are the ones suggested in the original paper (also see Appendix C.4) and the implementation is taken from their repository. Link.

## C.3   GAMMA DISTRIBUTION

Ali et al. (2000) suggest a method called *CVB* to simulate the run-time of a distributive network of computers. The method is based on several definitions:

**Definition 2.**  Task execution time variables:

- $\mu_{task}$ - mean time of tasks
- $V_{task}$ - variance of tasks
- $\mu_{mach}$ - mean computation power of machines
- $V_{mach}$ - variance of computation power of machines
- $\alpha_{task} = \frac{1}{V_{task}^2}$
- $\alpha_{mach} = \frac{1}{V_{mach}^2}$

$G(\alpha, \beta)$ is a random number generated using a gamma distribution, where $\alpha$ is the shape and $\beta$ is the scale.

For our case, all tasks are similar and run on a batch size of B. Therefore, the algorithm for deciding the execution-time of every task on a certain machine is reduced to one of the following:

---
**Algorithm 12** Task execution time - homogeneous machines
---
$\beta_{task} = \frac{\mu_{task}}{\alpha_{task}}$
$q = G(\alpha_{task}, \beta_{task})$
$\beta_{mach} = \frac{q}{\alpha_{mach}}$
for i from 0 to $K - 1$:
    $time = G(\alpha_{mach}, \beta_{mach})$

---

---
**Algorithm 13** Task execution time - heterogeneous machines
---
$\beta_{mach} = \frac{\mu_{mach}}{\alpha_{mach}}$
for j from 0 to $M$:
    $p[j] = G(\alpha_{mach}, \beta_{mach})$
$\beta_{task}[j] = \frac{p[j]}{\alpha_{task}}$
for i from 0 to $K - 1$:
    $time = G(\alpha_{task}, \beta_{task}[curr])$

---

where $K$ is the total amount of tasks of all the machines combined (the total number of batch iterations), $M$ is the total number of machines (workers), and $curr$ is the machine currently about to run.

We note that Algorithms 12 and 13 naturally give rise to stragglers. In the homogeneous algorithm, all workers have the same mean execution time but some tasks can still be very slow; this generally means that in every epoch a different machine will be the slowest. In the heterogeneous algorithm, every machine has a different mean execution time throughout the training. We further note that $p[j]$ is the mean execution time of machine $j$ on the average task.

In our experiments, we simulated execution times using the following parameters as suggested by Ali et al. (2000): $\mu_{task} = \mu_{mach} = B \cdot V_{mach}^2$, where $B$ is the batch size, yielding a mean execution time of $\mu$ simulated time units, which is proportionate to $B$. In the homogeneous setting $V_{mach} = 0.1$, whereas in the heterogeneous setting $V_{mach} = 0.6$. For both settings, $V_{task} = 0.1$.

Figure 3 illustrates the differences between the homogeneous and heterogeneous gamma-distribution. Both environments have the same mean (128) but the probability of having an iteration that is at

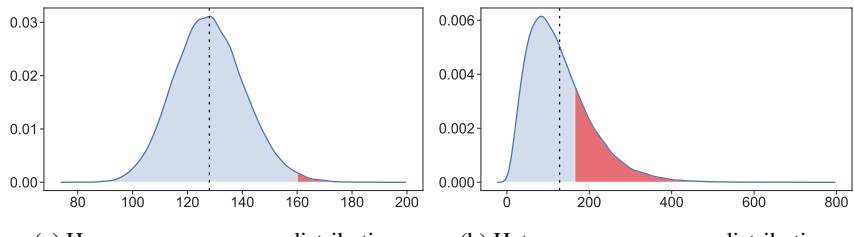

(a) Homogeneous gamma-distribution    (b) Heterogeneous gamma-distribution

Figure 3: Gamma-distribution in homogeneous and heterogeneous environments. The *x-axis* is the simulated time units the iteration takes while the *y-axis* is the probability. Both environments have the same mean (128 time units). The red area represents the probability to have an iteration which takes more than 1.25x longer than the mean iteration time.

least 1.25x longer than the mean (which means 160 or more) is significantly higher in the heterogeneous environment (27.9% in heterogeneous environment as opposed to 1% in the homogeneous environment).

## C.4 HYPERPARAMETERS

Since one of our intentions was to verify that penalizing the gradients linearly to the *Gap* is the factor that leads to a better final test error and convergence rate, we used the same hyperparameters across all algorithms tested. These hyperparameters are the original hyperparameters of the respective neural network architecture, which are tuned for a single worker.

**CIFAR-10 ResNet-20**

- Initial Learning Rate $\eta$: 0.1
- Momentum Coefficient $\gamma$: 0.9 with NAG
- Dampening: 0 (no dampening)
- Batch Size $B$: 128
- Weight Decay: 0.0005
- Learning Rate Decay: 0.1
- Learning Rate Decay Schedule: Epochs 80 and 120
- Total Epochs: 160

We note that Krizhevsky (2012) originally suggested 0.0001 as the Weight Decay hyperparameter for this framework. However, when we tuned the Weight Decay on a single worker we found that 0.0005 results in the best final test accuracy (92.43% as opposed to 91.63%).

**CIFAR-10/100 Wide ResNet 16-4**

- Initial Learning Rate $\eta$: 0.1
- Momentum Coefficient $\gamma$: 0.9 with NAG
- Dampening: 0 (no dampening)
- Batch Size $B$: 128
- Weight Decay: 0.0005
- Learning Rate Decay: 0.2
- Learning Rate Decay Schedule: Epochs 60, 120 and 160
- Total Epochs: 200

**ImageNet ResNet-50**

- Initial Learning Rate $\eta$: 0.1
- Momentum Coefficient $\gamma$: 0.9 with NAG
- Dampening: 0 (no dampening)
- Batch Size $B$: 256
- Weight Decay: 0.0001
- Learning Rate Decay: 0.1

- Learning Rate Decay Schedule: Epochs 30 and 60
- Total Epochs: 90

**WikiText-103 Transformer-XL**

- Initial Learning Rate $\eta$: 0.00025
- Dropout: 0.1
- Dampening: 0 (no dampening)
- Batch Size $B$: 64
- First Moment Coefficient $\beta_1$: 0.9
- Second Moment Coefficient $\beta_2$: 0.999
- $\epsilon$: $10^{-8}$
- Learning Rate Decay: Cosine from 0.00025 to 0
- Gradient Clipping: 0.25
- Total Iteration Steps: $K = 200000$

**Learning Rate Warm-Up** In the early stages of training, the network changes rapidly, causing error spikes. For all algorithms, we follow the gradual warm-up approach (Goyal et al., 2017) to overcome this problem: we divide the initial learning rate by the number of workers $N$ and ramp it up linearly until it reaches its original value after 5 epochs. We also use momentum correction (Goyal et al., 2017) in all algorithms to stabilize training when the learning rate changes.

## C.5 WEIGHT DECAY

When GA is used with Weight Decay, the gradients contain a weight decay element which also needs to be divided by the Gap as a part of the gradient.

## C.6 C COEFFICIENT

In Definition 1, we explained that we use a coefficient $C$ to measure the gap. In Section 5.3, using the parameter-wise method, $C$ is calculated per-parameter. To calculate $C$ per-parameter, we used a weighted-average in a manner similar to the technique used in Adam (Kingma & Ba, 2015). The mechanism is described in Algorithm 14.

---

**Algorithm 14** C Coefficient Calculation

---

**Require:** $\eta_{max}$ (usually $\eta_{max} = \eta_1$)
**Require:** $\beta_1 \in [0, 1)$: exponential decay rates for the moment estimates
**Initialize:** $C \leftarrow \mathbf{0^d}, \quad m_0 \leftarrow 0$
For k = 1...K do:
    Receive gradient $g_k^i$ from worker $i$
    Calculate update step $v_{k+1} \leftarrow \gamma v_k + g_k^i$
    Update biased second moment estimate $m_k \leftarrow \beta_1 m_{k-1} + (1 - \beta_1)v_{k+1}^2$
    Compute bias-corrected second order moment estimate $\hat{m}_k \leftarrow \frac{m_k}{1-\beta_1^k}$
    Calculate Coefficient $C \leftarrow \eta_{max}\left(\sqrt{\hat{m}_k} + \epsilon\right)$

---

Where is the parameter-wise method, all operations are executed per-parameter. Throughout our experiments we used $\beta_1 = 0.999$ and $\epsilon = 10^-8$ as suggested by Kingma & Ba (2015).

## C.7 Tabled Results

Table 4: ResNet-20 CIFAR10 Final Test Accuracy (Baseline 92.43%)

| N | SA | DANA-SA | NAG-ASGD | GA | ASGD | DANA | DANA-GA |
|---|---|---|---|---|---|---|---|
| 4 | 91.63±0.16 | 91.82±0.17 | 91.43±0.37 | 91.78±0.13 | 90.5±0.06 | **92.13±0.08** | 92.06±0.24 |
| 8 | 90.63±0.14 | 90.99±0.13 | 87.68±0.35 | 91.21±0.32 | 90.28±0.21 | 91.97±0.17 | **92.13±0.14** |
| 12 | 89.73±0.27 | 90.27±0.39 | 10.0±0.0 | 90.12±0.29 | 90.02±0.31 | 91.2±0.28 | **91.87±0.24** |
| 16 | 88.93±0.37 | 89.48±0.06 | 10.0±0.0 | 89.59±0.39 | 89.82±0.29 | 89.68±0.41 | **91.98±0.15** |
| 20 | 88.21±0.32 | 88.97±0.23 | 10.0±0.0 | 89.03±0.28 | 89.55±0.13 | 88.06±0.23 | **91.62±0.17** |
| 24 | 87.23±0.44 | 88.4±0.25 | 10.0±0.0 | 88.75±0.4 | 89.35±0.2 | 67.72±7.46 | **91.46±0.17** |
| 28 | 86.31±0.28 | 88.0±0.27 | 10.0±0.0 | 88.18±0.18 | 88.5±0.12 | 30.72±19.74 | **91.45±0.11** |
| 32 | 85.59±0.15 | 87.64±0.26 | 10.0±0.0 | 87.92±0.27 | 88.3±0.37 | 27.6±14.7 | **91.15±0.32** |
| 40 | 83.57±0.5 | 86.75±0.25 | 10.0±0.0 | 86.48±0.1 | 86.61±0.54 | 22.82±13.59 | **90.94±0.2** |
| 48 | 80.82±0.46 | 85.6±0.28 | 10.0±0.0 | 85.0±0.66 | 83.6±0.69 | 10.0±0.0 | **90.68±0.33** |

Table 5: WideResNet 16-4 CIFAR10 Final Test Accuracy (Baseline 95.17%)

| N | SA | DANA-SA | NAG-ASGD | GA | ASGD | DANA | DANA-GA |
|---|---|---|---|---|---|---|---|
| 4 | 94.41±0.16 | 94.39±0.26 | 94.81±0.11 | 94.89±0.16 | 92.99±0.15 | 95.0±0.13 | **95.08±0.1** |
| 8 | 93.38±0.16 | 93.52±0.02 | 92.83±0.61 | 94.32±0.16 | 92.91±0.09 | 94.67±0.06 | **94.9±0.16** |
| 12 | 92.46±0.16 | 92.68±0.21 | 44.35±28.22 | 94.02±0.1 | 92.81±0.22 | 94.27±0.15 | **94.84±0.1** |
| 16 | 91.51±0.14 | 91.92±0.07 | 23.36±26.72 | 93.72±0.08 | 92.48±0.25 | 93.56±0.26 | **94.77±0.19** |
| 20 | 90.62±0.17 | 91.37±0.23 | 33.41±30.31 | 93.35±0.11 | 92.28±0.31 | 92.57±0.35 | **94.68±0.11** |
| 24 | 90.06±0.16 | 90.5±0.14 | 11.62±3.23 | 92.92±0.03 | 92.13±0.3 | 89.93±0.59 | **94.39±0.1** |
| 28 | 89.38±0.35 | 90.18±0.17 | 31.91±17.68 | 92.56±0.08 | 91.6±0.26 | 75.32±12.14 | **94.39±0.22** |
| 32 | 88.7±0.06 | 89.35±0.16 | 19.52±19.03 | 92.35±0.22 | 91.22±0.19 | 68.1±14.03 | **94.27±0.1** |
| 40 | 87.26±0.21 | 88.23±0.25 | 10.0±0.0 | 91.65±0.16 | 90.53±0.24 | 28.97±25.07 | **93.84±0.12** |
| 48 | 85.65±0.26 | 86.64±0.78 | 12.26±4.52 | 90.97±0.38 | 89.54±0.31 | 22.31±11.43 | **93.63±0.13** |

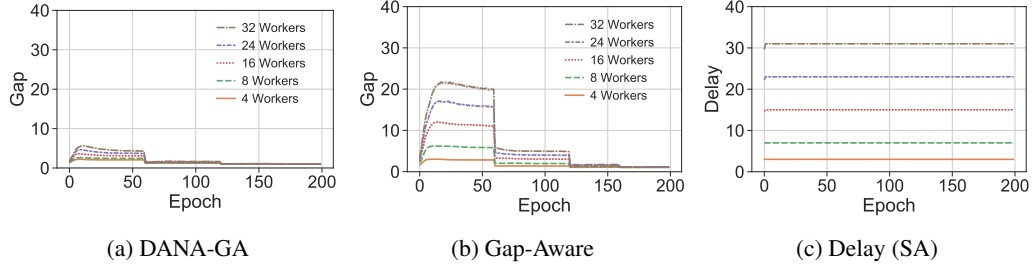

(a) DANA-GA       (b) Gap-Aware       (c) Delay (SA)

Figure 4: Delay and Gap throughout the training process for different number of workers using GA and DANA-GA. The figure shows the average (bold line) and standard deviation (band) of 5 runs for $N \in [4, 8, 16, 24, 32]$. All sub-figures are equally scaled to easily compare between them. GA penalizes the stale gradient much less than SA. DANA-GA penalizes the stale gradients even less than GA thanks to its approximation.

Table 6: WideResNet 16-4 CIFAR100 Final Test Accuracy (Baseline 76.72%)

| N | SA | DANA-SA | NAG-ASGD | GA | ASGD | DANA | DANA-GA |
|---|---|---|---|---|---|---|---|
| 4 | 74.74±0.3 | **76.69±0.19** | 76.27±0.2 | 75.64±0.08 | 72.42±0.31 | 76.21±0.22 | 75.75±0.11 |
| 8 | 73.14±0.16 | 75.66±0.19 | 74.24±0.27 | 74.32±0.31 | 72.8±0.25 | **76.03±0.13** | 75.64±0.26 |
| 12 | 71.66±0.23 | 74.73±0.38 | 69.29±0.56 | 73.86±0.3 | 72.34±0.28 | **75.72±0.27** | 75.41±0.2 |
| 16 | 70.39±0.27 | 73.76±0.19 | 67.37±0.74 | 72.82±0.17 | 71.99±0.3 | 75.0±0.26 | **75.01±0.17** |
| 20 | 69.51±0.23 | 72.3±0.33 | 37.98±7.21 | 72.34±0.2 | 71.63±0.18 | 73.41±0.4 | **74.75±0.22** |
| 24 | 68.66±0.1 | 70.5±0.44 | 9.67±4.89 | 71.74±0.17 | 71.15±0.34 | 71.26±0.49 | **74.76±0.16** |
| 28 | 67.48±0.3 | 67.67±0.3 | 6.35±7.41 | 71.22±0.34 | 70.58±0.36 | 68.7±1.25 | **74.41±0.45** |
| 32 | 65.67±0.41 | 63.52±0.77 | 12.71±7.69 | 70.64±0.31 | 69.91±0.26 | 66.73±1.15 | **74.33±0.28** |
| 40 | 62.31±0.34 | 66.06±0.23 | 9.56±5.1 | 69.75±0.4 | 69.25±0.48 | 64.29±1.1 | **73.61±0.26** |
| 48 | 58.78±0.57 | 64.46±0.27 | 4.24±3.15 | 68.34±0.19 | 67.63±0.43 | 27.42±10.95 | **73.33±0.23** |

## C.8 DELAY VS. GAP

To better illustrate that the delay is usually bigger than the Gap, we measured both sizes throughout the training process using different workers. The tested framework was CIFAR100 using WideResNet 16-4. We used the Gap-Aware and DANA-GA algorithms with the same hyperparameters described in Appendix C.4.

Figure 4 shows the average Gap and Delay ($\tau$) for every epoch throughout the training process. The Gap (Figure 4(b)) is constantly lower than the Delay (Figure 4(c)), especially towards the end of the training where the learning rate is small (which decreases the distance between the master's and worker's parameters). This figure illustrates how penalizing according to the Delay, as in Staleness-Aware, can easily over-penalize when the number of workers is large.

DANA uses an approximation of the master's parameters at the time of the future update, thus mitigating the staleness of the incoming gradients. This mitigation also reduces the Gap, as shown in Figure 4(a), which reduces the needed penalization when using DANA-GA.

## C.9 GAP VERSIONS

We tested three different Gap variations (See Section 5.3) to determine which technique has the best performance. The variations were tested on three different frameworks:

• CIFAR10 using ResNet-20
• CIFAR10 using WideResNet 16-4

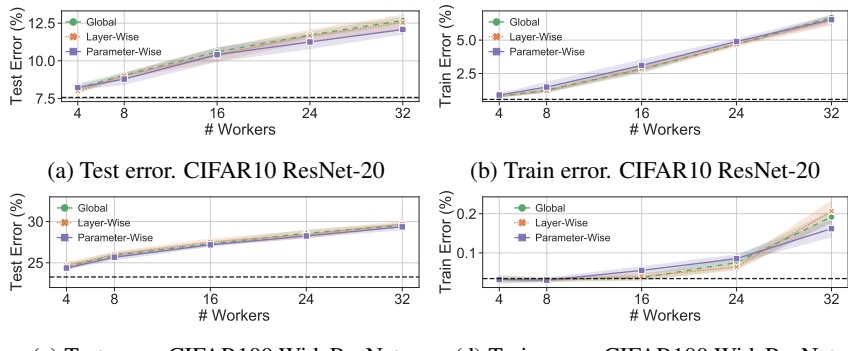

Figure 5: Final test and train error for different numbers of workers $N$. The figure shows the average (bold line) and standard deviation (band) of 5 runs. The black dashed line is the SGD error using a single worker.

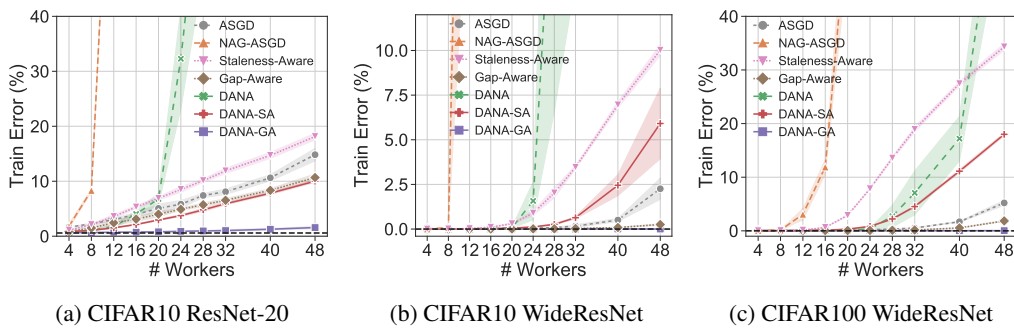

Figure 6: Final train error for different numbers of workers $N$. The figure shows the average (bold line) and standard deviation (band) of 5 runs on different frameworks.

• CIFAR100 using WideResNet 16-4

Each framework was trained on $N \in [4, 8, 16, 24, 32]$ asynchronous workers. Figure 5 contains the experiments not shown in Section 5.3 and demonstrates that *parameter-wise* Gap-Aware usually reaches better or similar final errors. The hyperparameters are the same ones detailed in Appendix C.4.

## C.10 CIFAR TRAINING

Figure 6 shows that DANA-GA always remains very close to the zero-error region. This means that, unlike other algorithms, DANA-GA is able to converge on the training set despite the gradient staleness. Figure 6 further demonstrates all of the concepts discussed in Section 6.1.

## C.11 CONVERGENCE RATE

Figure 7 shows that the convergence rate of GA is similar to that of SA, despite using a larger step size. This suggests that GA does not require more steps than SA to reach its (better) minima. DANA-GA's convergence rate remains very close to the baseline, which suggests that it doesn't require more steps to converge. This means that DANA-GA reaches a test error similar to the single worker case in the same number of steps, while enjoying asynchronous speedup.

Figure 8 demonstrates the same ideas discussed in the last paragraph on ImageNet using 64 asynchronous workers. However, since in this case the number of workers is much larger, both SA and DANA-SA perform very poorly due the the over-penalization of SA. This helps illustrate that GA is a better staleness mitigation method than SA. DANA-GA remains very close to the baseline despite the large number of workers used, demonstrating the superiority of DANA-GA.

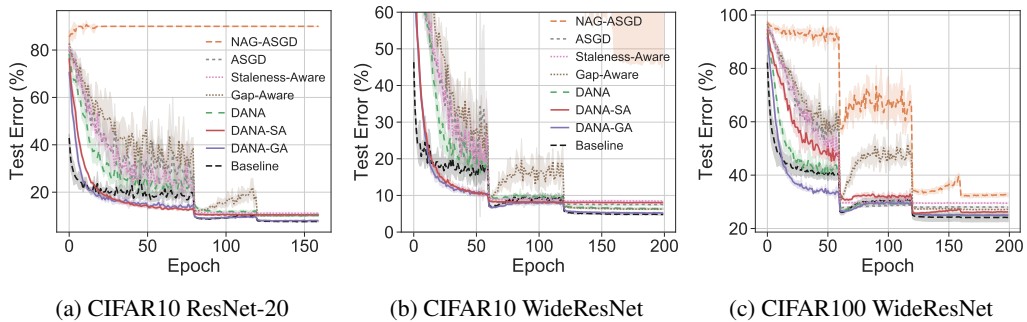

(a) CIFAR10 ResNet-20   (b) CIFAR10 WideResNet   (c) CIFAR100 WideResNet

Figure 7: Test error throughout the training using 16 workers. The figure shows the average (bold line) and standard deviation (band) of 5 runs on different frameworks.

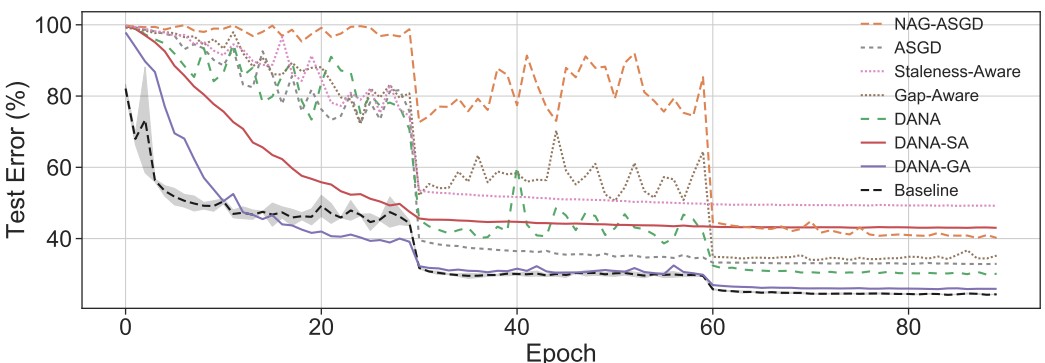

Figure 8: Test error throughout the training using 64 workers on ImageNet. DANA-GA remains close to the baseline. GA surpasses SA and DANA-SA.

## C.12 TUNED ASGD

We tuned the learning rate and momentum of ASGD on the CIFAR10 dataset with the ResNet-20 model using 32 workers. Tuning was performed using a grid search over 70 perturbations:

$$\eta \in [0.001, 0.003, 0.01, 0.03, 0.1, 0.3, 1]$$
$$\gamma \in [-0.9, -0.8, -0.5, -0.25, 0, 0.25, 0.5, 0.8, 0.9, 0.95]$$

As suggested by Mitliagkas et al. (2016), we also tested negative values of momentum to mitigate the implicit momentum created by the gradient staleness. Figures 9 to 11 show the results of the above experiments. The best final test error was given when:

- Figure 9: $(\eta = 0.03, \gamma = 0.5)$
- Figure 10: $(\eta = 0.3, \gamma = -0.8)$
- Figure 11: $(\eta = 0.3, \gamma = -0.5)$

This shows that the best hyperparameters can vary between frameworks. The best hyperparameters for a specific framework can also vary across different number of workers as all the hyperparameters found in this experiment are different from the best hyperparameters of the single worker. Tuning for the best hyperparameters for every different number of workers for each framework significantly increases the training time and is best avoided if possible.

DANA-GA surpasses the tuned ASGD in every framework, while DANA performs very poorly using 32 workers. This proves that gradient penalization is very beneficial to overcome gradient staleness and specifically that GA is a very successful gradient penalization method.

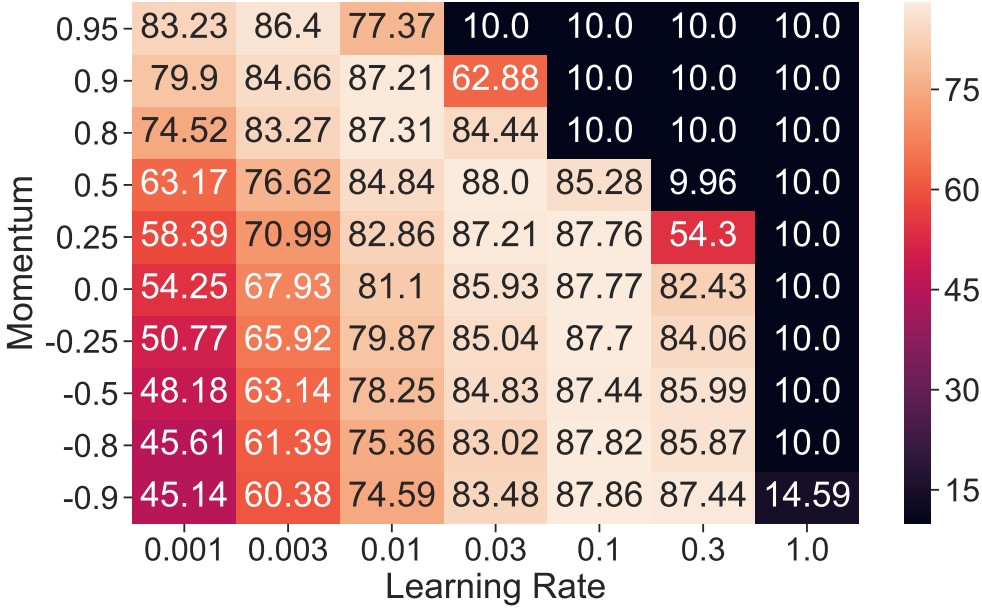

Figure 9: Test Accuracy of ASGD using 32 asynchronous workers on CIFAR10 ResNet-20 using different learning rate and momentum coefficients. The best accuracy achieved is 88% ($\eta = 0.03, \gamma = 0.5$). The tuning includes negative values of momentum

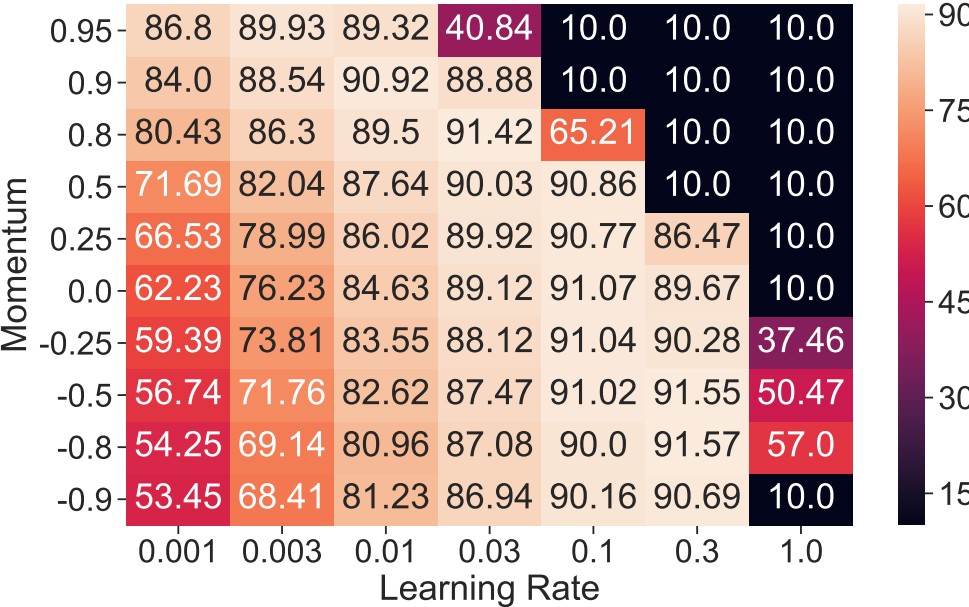

Figure 10: Test Accuracy of ASGD using 32 asynchronous workers on CIFAR10 WideResNet 16-4 using different learning rate and momentum coefficients. The best accuracy achieved is 91.57% ($\eta = 0.3, \gamma = -0.8$). The tuning includes negative values of momentum

## C.13 GRADIENT STALENESS NOISE

We notice that in the ImageNet experiments (Table 3) NAG-ASGD remains relatively close to the baseline even when the number of workers is large, as opposed to the CIFAR experiments, in which NAG-ASGD severely deteriorates as $N$ scales up. This phenomenon suggests that there is some

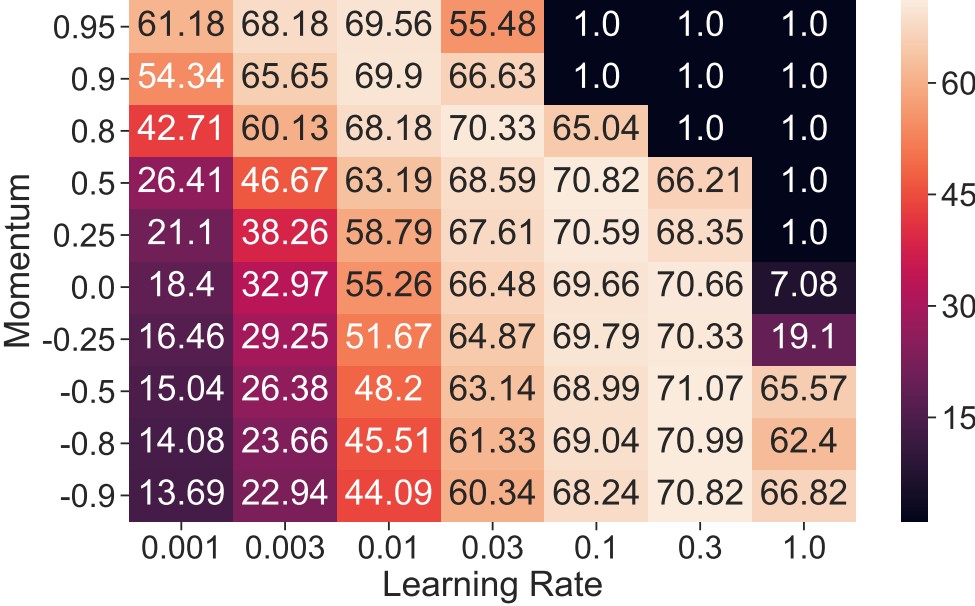

Figure 11: Test Accuracy of ASGD using 32 asynchronous workers on CIFAR10 WideResNet 16-4 using different learning rate and momentum coefficients. The best accuracy achieved is 71.07% ($\eta = 0.3, \gamma = -0.5$). The tuning includes negative values of momentum

*gradient staleness noise* that a framework can "tolerate" and still perform well. Following this intuition, it is reasonable that some gradient staleness should be allowed to go "un-penalized" to avoid limiting the step-size needlessly. This idea explains why SA and GA demonstrated relatively poor results in ImageNet, especially when the number of workers was relatively small. Though we consider this analysis beyond the scope of this work, it is relevant for this paper to note that we think that the tolerable *gradient staleness noise* depends on the size of the model and dataset, which suggests that GA can be further improved by correctly analysing the tolerable *gradient staleness noise* and starting the penalization accordingly. We plan to continue our research in this path as well.

## D ASYNCHRONOUS SPEEDUP

Cloud computing is becoming increasingly popular as a platform to perform distributed training of deep neural networks. Although synchronous SGD is currently the primary method (Mikami et al., 2018; Ying et al., 2018; Yamazaki et al., 2019; Goyal et al., 2017) used to distribute the learning process, it suffers from substantial slowdowns when run in non-dedicated environments such as the cloud. This shortcoming is magnified in heterogeneous environments and non-dedicated networks. ASGD addresses the SSGD drawback and enjoys linear speedup in terms of the number of workers in both heterogeneous and homogeneous environments even in non-dedicated networks. This makes ASGD a potentially better alternative for cloud computing.

Figure 12(a) shows the theoretically achievable speedup, based on the detailed gamma-distributed model, for asynchronous (GA and other ASGD variants) and synchronous algorithms using homogeneous and heterogeneous workers. The asynchronous algorithms can achieve linear speedup while the synchronous algorithm (SSGD) falls short as we increase the number of workers. This occurs because SSGD must wait in each iteration until all workers complete their batch. Figure 12(b) shows that ASGD-based algorithms (including GA, SA and DANA versions) are up to 21% faster than SSGD in homogeneous environments. In heterogeneous environments, ASGD methods can be 6x faster than SSGD. We note that this speedup is an underestimate, since our simulation includes only batch execution times. It does not model the execution time of barriers, all-gather operations, and other overheads which usually increase communication time, especially in SSGD.

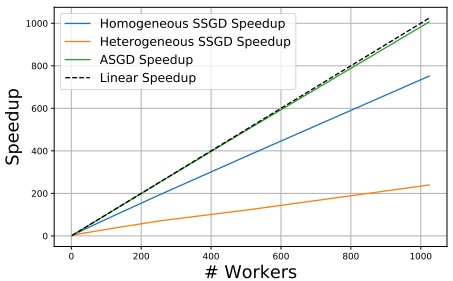

(a) Async (ASGD) and sync (SSGD) speedups.

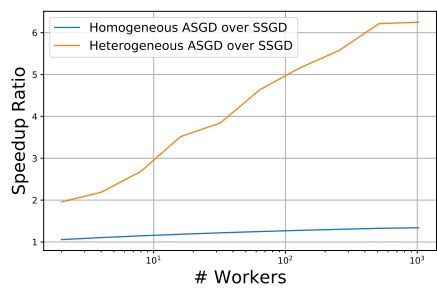

(b) ASGD speedup over SSGD. X axis is in log scale.

Figure 12: Theoretical speedups for any ASGD (such as GA, SA or DANA variants) and SSGD algorithms when batch execution times are drawn from a gamma distribution. Each line is an average of 20 runs with 100000 iterations per run. Communication overheads are not modeled; however, asynchronous algorithms are more communication efficient. Accounting for the communication overheads should expand the gap between the asynchronous and synchronous training.

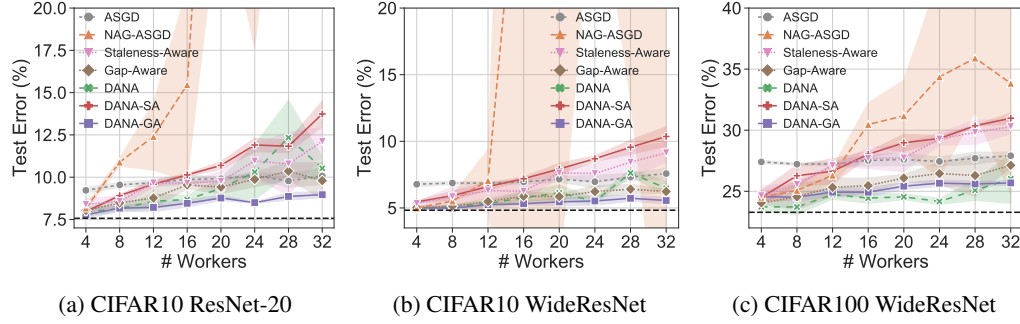

(a) CIFAR10 ResNet-20       (b) CIFAR10 WideResNet       (c) CIFAR100 WideResNet

Figure 13: Final test error for different numbers of heterogeneous workers $N$. The figure shows the average (bold line) and standard deviation (band) of 5 runs on different frameworks. The black dashed line represents the average result of SGD using a single worker.

## D.1   GAP-AWARE OVERHEAD

The Gap-Aware method requires the master to save the parameters of every worker and adds some simple calculations mentioned in Algorithm 4 and Algorithm 14. These calculations add no communication or computation requirements to the workers and introduce only a lightweight overhead to the parameter server compared to vanilla-ASGD. This lightweight overhead is similar to the overhead introduced by other ASGD-based methods such as Zheng et al. (2016). The space requirement is not critical since the master is usually implemented in a distributed manner, and the parameters are stored in the CPU-side memory, which is usually substantially larger than the total parameter size.

## D.2   HETEROGENEOUS EXPERIMENTS

We tested the performance of GA in reference to other algorithms when the asynchronous workers are heterogeneous. The setting was very similar to the one mentioned in Section 6, except that it this scenario we used the gamma-distribution to model heterogeneous workers (see Appendix C.3).

Figure 13 demonstrates that GA and DANA-GA are superior to the other tested algorithms in heterogeneous environments as well. When comparing between Figure 13 and Figure 2 it is noticeable that heterogeneous environments reach a higher accuracy. This is because in heterogeneous environments some workers are very fast compared to the other ones, thus their gradients are more accurate and arrive more frequently than the slow workers' gradients. Since in cloud computing, the workers can be either heterogeneous or homogeneous, we suggest using DANA-GA to maximize the results.

