# OpenReview forum: "Gap-Aware Mitigation of Gradient Staleness"
_ICLR.cc/2020/Conference — Accept (Poster)_

### Official Review · AnonReviewer1 · 2019-10-21
**Official Blind Review #1**

**Rating:** 3

**Review:**

This paper proposes a simple idea to mitigate gradient staleness in asynchronized distributed systems. Instead of scaling the staleness as in SA method, the authors propose to scale with the GAP, which is defined as the distance between current parameter and the staled parameter.

The paper is fairly well written, and the overall idea is interesting and simple in implementation. The authors also derive convergence bounds for the proposed algorithm and provide reasonable experimental results. I have the following comments:

1. I think the theory, at the current stage, is not sufficient. The authors only provide a convergence bound for the proposed method, which does not tell anything about the advantages over other methods such as the ASGD and SA methods. I think a detailed comparison should be conducted. From the current presentation, it seems there is no theoretical advantages of the proposed method over existing methods, which seems to suggest that the empirical advantage might not come from the algorithm itself.

I also notice that comparisons of convergence are illustrated in the appendix. However, it shows all SA and GA have similar convergence rates. This seems to indicate that SA and GA should performed similarly, which is not the case in the experiments.

2. For experimental results, in Figure 2 (and others), how do you get the errors for different workers? I mean, how long do you need to run for each case? You should make sure that all the settings should use the same computational resources for fair comparisons, but this is not described in the text, making the results doubtable.


**Experience Assessment:**

I have published one or two papers in this area.

**Review Assessment: Checking Correctness Of Derivations And Theory:**

I assessed the sensibility of the derivations and theory.

**Review Assessment: Checking Correctness Of Experiments:**

I assessed the sensibility of the experiments.

**Review Assessment: Thoroughness In Paper Reading:**

I read the paper at least twice and used my best judgement in assessing the paper.

---

> ### Author Response · Authors · 2019-11-06
> **Response for Reviewer #1**
>
> We thank Reviewer #1 for their time. We will address the issues raised in the review.
>
> 1)	The theoretical convergence rate shown in the paper is the convergence rate in terms of number of iteration steps. The asynchronous speedup, which is obtained by performing multiple iteration steps in parallel, is not a part of the theoretical convergence rate (but is shown in Appendix D). The theoretical convergence rate of GA-ASGD is proven to be the same as vanilla SGD, which proves that GA-ASGD achieves the best convergence rate possible. We note it is impossible for an asynchronous algorithm to have better convergence rate than SGD without improving the convergence rate of SGD as well. Furthermore, in the experiments section (specifically Figure 2) the only difference between the lines is the algorithm so this must be the reason for the improved results.
>
> As noted in Appendix C.11, though the convergence rates of GA and SA seem similar, the fact that GA used larger step sizes allows GA to reach a better final accuracy. To extend the intuition a little, it is a known result that if we were to perform two regular-SGD training procedures, one with a the optimal step size and one with a slightly smaller step size (both still converging), the smaller step size will seem to converge faster but to a lower accuracy since its search-space is limited.
>
> 2)	It is stated in the third line of Section 6 that for fair comparison “we used the same hyperparameters across all the tested algorithms”. These hyperparameters also include the number of epochs and the batch size which strictly determine the number of training steps (iterations). All hyperparameters are documented in Appendix C.4. We note that these hyperparameters remain constant for any number of workers which makes the comparison fair. For example, 2 workers working in parallel will each do about half the iterations a single worker would do in order to achieve the same fixed, predetermined, number of epochs. This means that across all runs, the total number of iterations stays the same.

---

### Official Review · AnonReviewer3 · 2019-10-23
**Official Blind Review #3**

**Rating:** 3

**Review:**

The paper introduces a new variant of asynchronous SGD, GA-ASGD, for distributed training. The goal is to mitigate the gradient staleness issue caused by asynchronously applying gradients to an old version of parameters. Prior work addresses this issue by penalizing the learning step of a worker linearly to its missed updates since getting a replica of parameters. However, that approach does not consider the differences between the old and new versions of parameters, which can cause over-penalization and under-penalization. The main contribution of this paper is to introduce a new way of measuring weight staleness and to explore the idea of penalizing the gradient itself to mitigate the staleness issue. The paper does a good job of discussing how GA can be applied to existing optimizers such as Adam. It performs empirical studies on ImageNet and Transformers-XL to conclude that GA-ASGD outperforms ASGD and the prior staleness aware approach. It also demonstrates the scalability of GA by scaling up to training with 128 asynchronous workers.

Strengths:
+ Introduced a novel approach to measure the parameter staleness, which helps penalize the gradients instead of the learning step.
+ The approach seems to be easily applicable without much additional hyperparameter tuning.
+ Evaluation on both image and NLP tasks demonstrate that GA leads to improvement over prior staleness aware approaches.

Weaknesses:
- Despite great improvements over prior ASGD based approaches, there is still a quite noticeable accuracy degradation compared to the SGD baseline.
- The evaluation is done with simulation, and no report of the number of training steps to converge and end-to-end training time, making it difficult to compare the efficiency with SGD based approaches.

Overall, I think this is good work. The idea of defining weights stableness as the minimal number of updates required to traverse the current distance between the master and worker weights seems to be reasonable. The comparison to prior ASGD based approaches are extensive, and the improvements seem decent. The fact that it does not require much or additional hyperparameter tuning also seems neat.

My major concern comes from results compared to SGD and motivation. The fact that almost all ASGD based workloads, even with the help of GA, cannot get on-par accuracy as the SGD baseline bothers me. In particular, the accuracy drops considerably as the number of asynchronous workers grows (Fig 1. and Fig. 2, and Table 3). For example,  the gap between the accuracy of GA vs. SGD can be as large as 3.46% when there are 128 workers. That gap might be closed with additional hyperparameter tuning, but it is unclear from the current draft and results. In the Transformer-XL example, GA has to limit the number of asynchronous workers to 4 in order to get close to baseline accuracy. It would have been better to show the trade-off between accuracy and performance in comparison with SGD.

On the other hand, the recent advance of SGD using large-batch training achieves great results on large model training such as BERT [1]. It helps to improve the compute/communication ratio, which seems to mitigate the straggler issue. Given that today's cloud service largely uses homogenous accelerators (TPU/GPU of the same SKU), it is less clear whether it is really beneficial to train with ASGD despite the improvements from GA.

Question:
One question is about how the batch size should be changed as the number of workers increases. The convergence analysis indicates that by increasing the batch size, the convergence speed of GA-ASGD will decrease. Does that mean the batch size should remain relatively unchanged to avoid negatively impacting the end-to-end training?

Is there any benefit to applying Gap to model parallelism paradigm such as pipeline parallelism?

[1] "Large Batch Optimization for Deep Learning: Training BERT in 76 minutes", by You et. al.

**Experience Assessment:**

I have read many papers in this area.

**Review Assessment: Checking Correctness Of Derivations And Theory:**

I assessed the sensibility of the derivations and theory.

**Review Assessment: Checking Correctness Of Experiments:**

I carefully checked the experiments.

**Review Assessment: Thoroughness In Paper Reading:**

I read the paper at least twice and used my best judgement in assessing the paper.

---

> ### Author Response · Authors · 2019-11-06
> **Response for Reviewer #3**
>
> We thank Reviewer #3 for their time and effort on this review. We will address the issues raised in the review.
>
> 1)	It is stated in the third line of Section 6 that for fair comparison “we used the same hyperparameters across all the tested algorithms”. These hyperparameters also include the number of epochs and the batch size which strictly determine the number of training steps. Both hyperparameters are documented in Appendix C.4.
>
> 2)	Regarding SSGD, our introduction states that large-batch training indeed achieves good accuracy. However, stragglers cause SSGD to slow down significantly. Though there are methods to mitigate the issue of the stragglers, this problem is still severe when training in non-dedicated environments, such as cloud computing, as mentioned in our paper and elaborated in Appendix D. For example, the paper [2] demonstrates in Figure 1 that even homogeneous GPUs on the cloud differ in iteration time from one another by more than 10x. This directly leads to a 10x slowdown when using large-batch training, which makes ASGD a very reasonable alternative.
> Other recent papers also acknowledge that ASGD methods are faster than large-batch methods [3-5]. This is especially true when the workers are heterogenous as discussed in Appendix D.1.
> Though the example of BERT [1], mentioned by the reviewer, reaches an impressive speedup of 49.1x, it was trained on a dedicated (and probably extremely expensive) network of hundreds of TPUs and not on the cloud. This sort of environment is out of reach for most users.
> Furthermore, as seen in Figure 2 in our paper, when the number of asynchronous workers is small, DANA-GA reaches the same accuracy as the baseline (which is optimal) even without hyperparameter tuning. This suggests that one may be able to enjoy some asynchronous speedup without hurting the accuracy by using few asynchronous workers, each worker using large batch training. This idea is beyond the scope of our work but it comes to show that ASGD methods still have merits.
>
> 3)	Regarding the first question, as mentioned in the paper, the convergence rate bound of Gap-Aware is similar to that of vanilla SGD. The observation regarding increasing the batch size to speed up the training might be intuitively correct, but it is relevant for regular SGD just the same. This makes the topic beyond the scope of this work.
>
> 4)	Regarding the second question, the answer is yes. Gap-Aware is fully compatible with model parallelism, especially synchronization schemes that introduce staleness such as PipeDream [6].
>
> [2] “Anytime Stochastic Gradient Descent: A Time to Hear from all the Workers” by Ferdinand et. al.
> [3] “Slow and stale gradients can win the race: Error-runtime trade-offs in distributed sgd” by Dutta et.al.
> [4] “Asynchronous Byzantine Machine Learning (the case of SGD)” by Damaskinos et. al.
> [5] “Asynchronous Stochastic Gradient Descent with Delay Compensation” by Zheng et. al.
> [6] “PipeDream: Fast and Efficient Pipeline Parallel DNN Training” by Harlap et. al.

---

> > ### Author Response · Authors · 2019-11-13
> > **Reaching baseline accuracy**
> >
> > Following the concerns of Reviewer #3, we've performed some hyperparameter tuning on CIFAR10, using the Resnet-20 model with 48 asynchronous workers using the DANA-GA algorithm. To relax the reviewer's concerns, we've found that using a batch size of 32 and a learning rate of 0.02 (the rest of the huperparameters are unchanged), reaches 92.19% test accuracy. This accuracy is very close to the baseline vanilla-SGD accuracy using a single worker (0.24% difference), which demonstrates that asynchronous SGD can indeed reach vanilla-SGD accuracy in the same number of epochs using DANA-GA and hyperparameter tuning.

---

> > > ### Comment · AnonReviewer3 · 2019-11-14
> > > **Thank you for updating the results**
> > >
> > > Thank you for providing the updated results, which I appreciate. However, the new result indicates that it indeed requires extra hyperparameter tuning to reach on-par accuracy, which seems to against one of the major claims of the paper:  GA can be used without re-tuning the hyperparameters. Furthermore, it is hard for me to assess the generality of the obtained results. Overall, one thing that might be helpful for further improving the paper is to include both final results without and with some additional but non-extensive hyperparameter tuning, so that it shows GA does not compromise convergence quality.

---

> > > > ### Author Response · Authors · 2019-11-14
> > > > **Thank you for your reply and suggestion**
> > > >
> > > > We thank you for the reply and appreciate your suggestion to improve our paper. There are two things we wish to emphasize regarding your remarks:
> > > > 1)	We never claimed that tuning is not beneficial when using GA so there is no contradiction in our paper. Obviously, tuning is always beneficial; however, the effort required for tuning may not be worth it when using GA, since GA already performs better than other ASGD-based algorithms and relatively close to vanilla-SGD, without tuning. We note that tuning has more potential to be beneficial when the number of workers is large.
> > > > 2)	Your major concern regarding our paper is based on the claim that ASGD (even with the help of GA) is always inferior to large-batch SGD (SSGD). Since our comments relieve your sole major concern, we expect your overall review to change. If this is not the case, we would very much like to know why in order to improve our paper.

---

### Official Review · AnonReviewer2 · 2019-10-25
**Official Blind Review #2**

**Rating:** 6

**Review:**

This paper studies training large machine learning models in a distributed setup. For such a setup, as the number of workers increases, employing synchronous stochastic gradient descent incurs a significant delay due to the presence of straggling workers. Using asynchronous methods should circumvent the issue of stragglers. However, these asynchronous methods suffer from the stale gradients where by the time a worker sends the gradients to the master server, the model parameters have changed based on the gradients received from other workers. This leads to severe performance degradation in the models trained by using asynchronous methods, especially where the number of workers scales.

This paper proposes a gap-aware method to reduce the adverse impact of stale gradient on the asynchronous distributed learning. In particular, when the master receives a gradient from a worker, it computes the norm of the difference between the current model parameter and the past model parameter associated with the gradient. The master then computes *gap* value based on this norm and the norm of the average gradient. Before employing the gradient, the master scales the gradient by the computed gap value. The paper establishes the convergence rate for the gap-aware asynchronous method which is similar to the convergence rate of SGD. The paper then performs an extensive experimental evaluation of the proposed method over different datasets and models. The empirical results demonstrate the advantage of the proposed gap-aware method over other baselines.

Pros

- The extensive empirical evaluation shows that the proposed method is effective in preventing performance degradation in an asynchronous setup across tasks and models.

- The method outperforms other solutions to combat stale gradient, such as the staleness-aware method by Zhang et al.

- The paper shows that the proposed method can be combined with DANA (Hakimi et al.) and achieves the performance very close to the synchronous setting while realizing the speed up provided by the asynchronous methods.

Cons

- It was not clear to the reviewer how the convergence analysis of the proposed method differs from the existing analysis in the literature and if any novel ideas were involved in obtaining the theoretical results presented in the paper.

- The gap-aware method increases the overhead at the master as it needs to store the most recent model parameters sent to each of the workers. This is much higher than the staleness-aware method where the master stores a single scalar for each worker. The paper barely discusses such overheads associated with the proposed method.

**Experience Assessment:**

I have read many papers in this area.

**Review Assessment: Checking Correctness Of Derivations And Theory:**

I carefully checked the derivations and theory.

**Review Assessment: Checking Correctness Of Experiments:**

I carefully checked the experiments.

**Review Assessment: Thoroughness In Paper Reading:**

I read the paper at least twice and used my best judgement in assessing the paper.

---

> ### Author Response · Authors · 2019-11-06
> **Response for Reviewer #2**
>
> We thank Reviewer #2 for their time and effort on this review. We will address the issues raised in the review.
>
> 1)	In the theoretical convergence analysis, we prove that the convergence rate of Gap-Aware, in terms of iteration steps, is as good as the convergence rate of vanilla SGD (despite performing these steps in parallel). This means that theoretically, Gap-Aware achieves the best convergence rate possible. The purpose of the convergence analysis was to prove that even in theory, Gap-Aware’s convergence bounds are at least as good as those of SA or even vanilla SGD. The paper’s novelty is introducing the Gap-Aware method and empirically showing it converges to better accuracy than SA or other methods in a convergence rate that doesn’t fall from that of SGD. This novelty is not diminished if the methods used in the theoretical convergence analysis are not unique.
>
> 2)	The reviewer is correct in stating that Gap-Aware introduces an overhead at the master; however, this is common for many ASGD-based algorithms since these operations are negligible when using parameter server optimizations ([1-6] provide examples of such optimizations). To alleviate the reviewer’s concerns, we note that simply storing parameters in the master can be done by a simple CPU and the time it takes to perform the computation is negligible, especially when compared to the time a gradient iteration takes. We will add a section discussing this issue in the Appendix.
>
> 3) We would like to clarify a possible misunderstanding. In the Pros. section, the reviewer stated that DANA-GA achieves performance which is very close to the synchronous setting; However, the dashed line which appears in Figure 2 (and other places along the paper) represents the error of a single worker. This means that this is the result of vanilla-SGD and not synchronous-SGD (synchronous-SGD might experience a degradation of performance due to the large batch-size)). Ultimately, the performace of synchronous-SGD has not been compared to in this paper and DANA-GA may surpass it in some cases.
>
> [1] “Large scale distributed deep networks” by Dean et. al.
> [2] “Priority-based parameter propagation for distributed dnn training” by Jayarajan et. al.
> [3] “Adaptive communication strategies to achieve the best error-runtime trade-off in local update sgd” by Wang et. al.
> [4] “Tictac: Accelerating distributed deep learning with communication scheduling” by Hashemi et. al.
> [5] “Beyond data and model parallelism for deep neural networks” by Jia et.al.
> [6] “3lc: Lightweight and effective traffic compression for distributed machine learning” by Lim et. al.

---

### Decision · Program_Chairs · 2019-12-19

**Decision:**

Accept (Poster)

**Comment:**

The authors propose a novel approach for measuring gradient staleness and use this measure to penalize stale gradients in an asynchronous stochastic gradient set up. Following previous work, they provide a convergence proof for their approach. Most importantly, they provide extensive evaluations comparing against previous approaches and show impressive gains over previous work.

After the author response, the primary concerns from reviewers is regarding the gap between the proposed method and single worker SGD/synchronous SGD. I feel that the authors have made compelling arguments that ASGD is an important optimization paradigm to consider, so their improvements in narrowing the gap are of interest to the community. There were some concerns about the novelty of the theory, and my impression is that theorem is straightforward to prove based on assumptions and previous work, however, I view the main contribution of the paper as empirical.

This paper is borderline, but I think the impressive empirical results over existing work on ASGD is a worthwhile contribution and others will find it interesting, so I am recommending acceptance.